

# Seasonal dynamics of the COS and CO$_2$ exchange of a managed temperate grassland

Felix M. Spielmann[1], Albin Hammerle[1], Florian Kitz[1], Katharina Gerdel[1], Georg Wohlfahrt[1]

[1]Department of Ecology, University of Innsbruck, Innsbruck, 6020, Austria

*Correspondence to*: Georg Wohlfahrt (Georg.Wohlfahrt@uibk.ac.at)

**Abstract.** Gross primary productivity (GPP), the CO$_2$ uptake by means of photosynthesis, cannot be measured directly on ecosystem scale, but has to be inferred from proxies or models. One newly emerged proxy is the trace gas carbonyl sulfide (COS). COS diffuses into plant leaves in a fashion very similar to CO$_2$, but is generally not emitted by plants. Laboratory studies on leaf level gas exchange have shown promising correlations between the leaf relative uptake (LRU) of COS to CO$_2$ under controlled conditions. However, *in situ* measurements including daily to seasonal environmental changes are required, to test the applicability of COS as a tracer for GPP at larger temporal scales. To this end, we conducted concurrent ecosystem scale CO$_2$ and COS flux measurements above an agriculturally managed temperate mountain grassland. We also determined the magnitude and variability of the soil COS exchange, which can affect the LRU on ecosystem level. The cutting and removal of the grass at the site had a major influence on the soil as well as the total exchange of COS. The grassland acted as a major sink for CO$_2$ and COS during periods of high leaf area. The sink strength decreased after the cuts and the grassland turned into a net source for CO$_2$ and COS on ecosystem level. The soil acted as a small sink for COS when the canopy was undisturbed, but also turned into a source after the cuts, which we linked to higher incident radiation hitting the soil surface. However, the soil contribution was not large enough to explain the COS emission on ecosystem level, hinting to an unknown COS source possibly related to dead plant matter degradation. Over the course of the season, we observed a concurrent decrease of CO$_2$ and COS uptake on ecosystem level. With the exception of the short periods after the cuts, the LRU under high light conditions was rather stable and indicates a high correlation between the COS flux and GPP across the growing season.

## 1 Introduction

Carbonyl sulfide (COS) is the most abundant sulfur-containing gas in the atmosphere with tropospheric mixing ratios of ~500 ppt. Within the atmosphere, COS acts as a greenhouse gas with a 724 times higher direct radiative forcing efficiency as CO$_2$ (Brühl et al., 2012). After reaching the stratosphere, it reacts to sulfur aerosols via oxidation and photolysis, hence contributing to the backscattering of solar radiation and having a cooling effect on Earth's atmosphere (Krysztofiak et al., 2015;Whelan et al., 2018). The intra-seasonal atmospheric COS mixing ratio follows the pattern of CO$_2$ as terrestrial vegetation acts as the largest known sink for both species (Montzka et al., 2007;Whelan et al., 2018;Le Quere et al., 2018). However, the summer drawdown for COS is 6 times stronger than for CO$_2$ (Montzka et al., 2007) as COS is generally not emitted by plants like CO$_2$, which is released in respiration processes.

The uptake of COS by plants is mostly mediated by the enzyme carbonic anhydrase (CA), but also photolytic enzymes like Ribulose-1,5-bisphosphate-carboxylase/-oxygenase (Rubisco) (Lorimer and Pierce, 1989). This in turn means that COS and CO$_2$ share a similar pathway into leaves through the boundary layer, the stomata and the cytosol, up to their reaction sites. Compared to CO$_2$, COS is processed in a one-way reaction to H$_2$S and CO$_2$ (Protoschill-Krebs and Kesselmeier, 1992;Notni et al., 2007) and therefore not released by plants (with the exception of severely stressed plants (Bloem et al., 2012;Gimeno et al., 2017)). That makes COS an interesting tracer for estimating the stomatal conductance and the gross uptake of CO$_2$,



referred to as gross primary production (GPP), on ecosystem level (Asaf et al., 2013;Kooijmans et al., 2017;Kooijmans et al.,
2019). However, to estimate GPP using COS, the relative uptake of COS to GPP deposition velocities (LRU) must be known
beforehand (see Eq.1), so that GPP can be estimated on the basis of the COS flux.
$$LRU = \frac{F_{COS}}{\chi_{COS}} \Big/ \frac{F_{CO_2}}{\chi_{CO_2}}$$ (Eq.1)
$F_{COS}$ is the COS leaf flux (pmol m$^{-2}$ s$^{-1}$), $F_{CO_2}$ is the gross $CO_2$ uptake on leaf level (µmol m$^{-2}$ s$^{-1}$) and $\chi_{COS}$ and $\chi_{CO_2}$ are the
ambient COS and $CO_2$ mixing ratios in ppt and ppm, respectively. Leaf level studies for $C_3$ plants have estimated the LRU to
be around 1.7 with the 95% confidence interval between 0.7 and 6.2 (Whelan et al., 2018;Seibt et al., 2010;Sandoval-Soto et
al., 2005). The large spread of the LRU most likely originates from differences between plant species, for example, leaf
structure and plant metabolism (Wohlfahrt et al., 2012;Seibt et al., 2010), which questions the applicability of the concept of
LRU in real-world ecosystems under naturally varying environmental conditions. It is also known that the LRU is just stable
under high light conditions, since the uptake of $CO_2$ by means of photosynthesis is a light driven process, while CA is able to
process COS independently of light conditions (Maseyk et al., 2014;Yang et al., 2018;Stimler et al., 2011). Any model of
LRU should therefore reflect diurnal changes in light conditions. Kooijmans et al. (2019) recently discovered that the vapor
pressure deficit (VPD) appears to have a stronger control on $F_{COS}$ than on $F_{CO_2}$, in an evergreen needle forest. If generally
true, this would add further variability to the LRU and complicating the application of COS to estimate GPP. Besides inter-
specific differences in LRU, the question remains unanswered if the LRU is also susceptible to seasonal changes of
ecosystems for example, changes in species composition or phenology, which would further complicate the application of
COS in carbon cycle research. Maseyk et al. (2014) observed COS emissions on ecosystem scale over a winter wheat field
going into senescence, indicating that potentially strong sources of COS could distort LRU.
Since CA and other enzymes known to emit or take up COS are also present in microorganisms (Ogawa et al., 2013;Seefeldt
et al., 1995;Ensign, 1995;Smeulders et al., 2013;Whelan et al., 2018), recent studies have also quantified the contribution of
soils to the COS ecosystem flux (Kooijmans et al., 2017;Spielmann et al., 2019;Maseyk et al., 2014). COS soil fluxes could
modify the LRU on ecosystem level and hence inferred GPP, if they are substantial compared to COS canopy fluxes. Similar
to the ecosystem fluxes, the soil fluxes could not only be prone to diurnal, but also seasonal changes, depending on the
substrate availability, environmental conditions (e.g. soil temperature and moisture), substrate quality and quantity, and
changes in composition of the microbial communities (Kitz et al., 2019;Meredith et al., 2019). Recent studies have also
linked COS soil emissions to abiotic processes dependent on light and/or temperature (Whelan and Rhew, 2015;Kitz et al.,
2019;Meredith et al., 2018).
The goal of our study was to provide new insights into the seasonal variability of COS fluxes on ecosystem, soil and canopy
level. To this end, we conducted a 6-month campaign on a managed temperate mountain grassland, measuring ecosystem as
well as soil COS fluxes. Since the grassland was cut four times during the campaign, we were able to observe multiple
growing cycles and investigate the diel and seasonal changes of the COS fluxes and the LRU in this highly dynamic
ecosystem. We hypothesize that (H1) the grassland, given its large $CO_2$ uptake capacity (Wohlfahrt et al. 2008), is a major
sink for COS and that the sink strength decreases over the course of the season, (H2) the drying of the cut grass leads to a
release of COS, (H3) the LRU will change after the cuts, due to stressed plants and drying plant parts in the field, but is
otherwise stable,  (H4) the cuts turn the soil into a COS source, due to the larger amount of light reaching the soil surface
(Kitz et al., 2017), but once a reasonably high leaf area index (LAI) has developed, COS is taken up by soil.



## 2 Methods

### 2.1 Study site and period

The study was conducted at an intensively managed mountain grassland in the municipal territory of Neustift (Austria) in Stubai valley (FLUXNET ID: AT-Neu; doi: 10.18140/FLX/1440121). The grassland is situated at an elevation of 970 m a.s.l. in the middle of the flat valley bottom. The soil was classified as Fluvisol with an estimated depth of 1 m with the majority of roots located within the first 10 cm. Measurements were conducted between 01.05.2015 and 31.10.2015 (183 days). The vegetation was described as Pastincao-Arrhenatheretum and consisted mainly of *Dactylis glomerata, Festuca pratensis, Alopecurus pratensis, Trisetum flavescens, Ranunculus acris, Taraxacum offcinale, Trifolium repens, Trifolium pratense,* and *Carum carvi (Kitz et al., 2017)*. During the campaign, the grassland was cut four times (02.06./07.07./21.08./01.10.2015) and the biomass left to dry on the field for up to one day, before being removed as silage. The field site was fertilized with organic manure at the end of the season (07.10.2015).

### 2.2 Leaf area index

The LAI was estimated from assessments of the average canopy height, which were related to destructive LAI measurements, using the following sigmoid function:

$$LAI = 1 \Big/ \left(1 + \exp\left(-(a_1 DOY + a_2)\right)\right)(b_1 - b_2) \tag{Eq.2}$$

where DOY is the day of the year and a1, a2, b1 and b2 are factors that were optimized for each growing period, for example, before the first cut, between cuts and after fourth cut (Wohlfahrt et al., 2008). Additionally, biomass samples were taken at 15 occasions, to assist with the LAI calculation.

### 2.3 Mixing ratio measurements

The $CO_2$ ($\chi_{CO2}$) and COS ($\chi_{COS}$) mixing ratios were measured using a Quantum Cascade Laser (QCL) Mini Monitor (Aerodyne Research, Billerica, MA, USA) at a wavenumber of ca. 2056 $cm^{-1}$ and at a frequency of 10 Hz. To minimize the effect of air temperature ($T_{air}$) changes on the instrument, we placed it in an insulated box which in turn was located in a climate controlled instrument hut (30°C). The cooling of the laser was achieved by a chiller (ThermoCube 400, Solid State Cooling Systems, Wappinger Falls, NY, USA).

We used ¼ inch Teflon™ tubing, stainless steel fittings (SWAGELOK, Solon, OH, USA and FITOK, Offenbach, HE, Germany), Teflon Filters (Savilex, EdenPrarie, MN, USA) as well as COS-inert valves (Parker-Hannafin, Cleveland, OH, USA) to ensure that only materials known not to interact with COS were used for the measurement and calibration airflow. The $H_2O$ and $CO_2$ mixing ratios ($\chi_{H2O}$ & $\chi_{CO2}$) were measured by a closed-path infrared gas analyzer (IRGA) (Licor 6262, LICOR Biosciences, Lincoln, NE, USA). Since the data of the QCL, the sonic anemometer and the IRGA were saved on two separate PCs, a network time protocol software (NTP, Meinberg, NI, Germany) was used to keep the time on both devices synchronized. We corrected known $\chi_{COS}$ drift issues of the QCL (Kooijmans et al., 2016) by doing half hourly calibrations for 1 min with a gas of known $\chi_{COS}$. The gas cylinders (working standards) used for the calibrations were either pressurized air (UN 1002) or nitrogen (UN 1066), which were cross-compared (when working standard cylinders were full and close to empty) to an Aculife-treated aluminum pressurized air cylinder obtained from the National Oceanic & Atmospheric Administration (NOAA). The latter was analyzed by the central calibration laboratory of NOAA for its $\chi_{COS}$ using gas chromatography on 06.04.2015. We then linearly interpolated between the offsets of the half hourly calibrations and used the retrieved values to correct the high frequency COS data. Due to issues with the scale gas cylinder, no absolute concentrations were available before the first cut and therefore no LRU was calculated for this period.



### 2.3.1 Mixing ratio measurements within the canopy

In order to investigate the $\chi_{COS}$ within the canopy, we used a multiplexer and 8 ¼ inch Teflon™ tubes to measure the $\chi_{COS}$ at 8 heights within and above the canopy i.e. at 2, 5, 10, 20, 30, 40, 50 & 250 cm height above ground with a tube length of 15 m for each height. The upper two intakes were located at the eddy covariance measurement and canopy height, respectively. Each height was measured for 1 min at 1 Hz and 2 l min$^{-1}$, while the other lines were each flushed at 2 l min$^{-1}$. The $\chi_{COS}$ drift was also corrected by doing half hourly calibrations (see section 2.3).

### 2.4 COS soil fluxes

### 2.4.1 Soil chamber setup

To quantify soil COS fluxes, we installed four stainless steel (SAE grade: 316L) rings 5 cm into the soil. They remained on site for 112 days (10.06.2015 – 30.09.2015). Two additional rings were installed on the 31.08.2015 and the 02.10.2015 to examine any long-term effects of the ring placement and to replace the original rings for the measurements in September and October. The aboveground biomass within each ring was removed at least one day prior to each measurement day. The roots within as well as the vegetation surrounding the rings were not removed and natural litter was left in place. At days without measurements the soil within the rings was covered by fleece to prevent it from drying out.

To measure the soil fluxes, a transparent fused silica-glass chamber (Kitz et al., 2017) was placed into the water filled channel of the steel rings, while air was sucked through the chamber to the QCL at a flow rate of 1.5 l min$^{-1}$. The chamber $\chi_{COS}$ was then compared with the ambient $\chi_{COS}$ above the chamber, using a second inlet to which we switched before the chamber measurement and after reaching stable readings inside the chamber. Overall, 243 chamber measurements were conducted over the course of the campaign including day and nighttime measurements. Additional manual measurements included a hand-held sensor (WET-2, Delta-T Devices, Cambridge, England) to measure soil water content (SWC) and soil temperature ($T_{soil}$) at a soil depth of 5 cm simultaneously with the soil chamber measurements next to the rings.

### 2.4.1 COS soil flux calculation

The COS soil flux was calculated using the following equation:

$$F = {q(\chi_{cos2} - \chi_{cos1})}/{A} \qquad \text{(Eq.3)}$$

where F is the COS soil flux (pmol m$^{-2}$ s$^{-1}$), q denotes the flowrate in (mol s$^{-1}$), $\chi_{COS2}$ and $\chi_{COS1}$ are the chamber and ambient $\chi_{COS}$ in ppt, respectively and A the soil surface area (0.032 m$^2$) covered by the chamber. A more detailed description can be found in Kitz et al. (2017).

### 2.4.2 COS soil exchange modelling

Due to the removal of the aboveground biomass and the consequent higher shortwave radiation reaching the soil surface in the chambers, compared to the soil below the canopy, we simulated the soil COS exchange for natural conditions. The soil flux was modelled using our measured soil fluxes and additionally retrieved soil and meteorological data - $T_{soil}$, soil water content (SWC) at 5 cm depth next to the chambers and incident shortwave radiation reaching the soil surface ($R_{SW-soil}$) - as input for a random forest regression model (Liaw and Wiener, 2002). The soil fluxes were modelled on half hourly basis for the whole duration of the measurement campaign to calculate the COS canopy fluxes from the difference of the COS ecosystem and soil fluxes. To this end we used the scikit-learn (sklearn Ver. 0.19.1) package, the pandas library and the Python Software Distribution Anaconda (Ver. 5.2.0) in the command shell Ipython (Ver. 6.4.0) based on the Programming language Python (Ver. 3.3.5). We used the Beer-Lambert law to model $R_{SW\_soil}$ under undisturbed conditions as the aboveground vegetation was removed to measure the COS exchange of bare soil:





$R_{SW-soil} = R_{SW}\exp(-K\ LAI)$ (Eq.4)
where $R_{SW\text{-}soil}$ (Wm$^{-2}$s$^{-1}$) is the shortwave radiation (SW) reaching the soil surface, $R_{SW}$ is the incoming SW radiation
reaching the top of the canopy, LAI is the plant area index (Eq. 2) and K is the canopy extinction coefficient assuming a
spherical leaf inclination distribution (Wohlfahrt et al., 2001), which was calculated using the following equation:
$K = \frac{1}{2\cos(\psi)}$ (Eq.5)
where $\psi$ is the zenith angle of the sun in radians.

A random forest with 1000 trees was grown which resulted in an out of bag (OOB) score of (0.82). The optimal input
parameters, including maximum tree depth, were determined with the function GridSearchCV from the sklearn package.

### 162 2.5 Ecosystem fluxes

### 163 2.5.1 Setup for ecosystem fluxes

The COS, $CO_2$ and $H_2O$ ecosystem fluxes were obtained using the eddy covariance method (Aubinet et al., 1999;Baldocchi,
2014). We used a 3-axis sonic anemometer (Gill R3IA, Gill Instruments Limited, Lymington, UK) to obtain-high resolution
data of the 3 wind components. The intake of the tube for the eddy covariance measurements was installed in close
proximity to the sonic anemometer and insulated as well as heated above $T_{air}$ to prevent condensation within the tube. The air
was sucked to the QCL at a flowrate of 7 l min$^{-1}$ using a Vacuum Pump (Agilent Technologies, CA, USA).

### 169 2.5.2 Ecosystem flux calculation

In a first step we used a self-developed software to determine the time lag, introduced by the separation of tube intake and
the sonic anemometer and the tube length, between the QCL and sonic dataset (Hortnagl et al., 2010). The data were then
processed using the software EdiRe (University of Edinburgh, UK) and Matlab2019a (MathWorks, MA, USA). We used the
laser drift corrected $\chi_{COS}$ data and linear detrending to process the data before following the procedure to correct for sensor
response, tube attenuation, path averaging and sensor separation following Gerdel et al. (2017). The random flux uncertainty
was calculated following Langford et al. (2015).
We estimated the COS canopy flux from the difference between the measured COS ecosystem and the modelled COS soil
flux.

### 178 2.5.3 Flux partitioning and leaf relative uptake

The GPP on ecosystem level was determined using the FP+ model put forward by Spielmann et al. (2019). The model
estimates the GPP on the basis of nighttime net ecosystem exchange (NEE) measurements of $CO_2$ that are assumed to
provide the temperature response of the ecosystem respiration (RECO) as well as a light dependency curve to estimate GPP
based on the daytime NEE (Lasslop et al., 2010):
$NEE = \frac{\alpha\beta R_{PAR}}{\alpha R_{PAR}+\beta} + rb\ e^{E_0(\frac{1}{T_{ref}-T_0}-\frac{1}{T_{air}-T_0})}$ (Eq.6)
where $\alpha$ denotes the canopy light utilization efficiency ($\mu$mol $CO_2$ $\mu$mol$^{-1}$ photons), $\beta$ the maximum $CO_2$ uptake rate of the
canopy at light saturation ($\mu$mol $CO_2$ m$^{-2}$ s$^{-1}$), $R_{PAR}$ the incoming photosynthetic active radiation ($\mu$mol m$^{-2}$ s$^{-1}$), rb the
ecosystem base respiration ($\mu$mol m$^{-2}$ s$^{-1}$) at the reference temperature $T_{Ref}$ (°C), which is set to 15°C, $T_{air}$ (°C) refers to the
air temperature and $E_0$ (°C) to the temperature sensitivity of RECO. $T_0$ was kept constant at -46.02°C. We did not use the
VPD modifier of beta put forward by Lasslop et al. (2010) as its value could not be estimated with confidence.
The FP+ model by Spielmann et al. (2019) extends the daytime FP (Eq.6) to also estimate the COS ecosystem fluxes by
linking the GPP resulting from the first part on the right-hand side of Eq.6 with the COS exchange through:



$$F_{COSmodel} = \frac{GPP\,LRU}{\chi_{CO2}} \Big/ \chi_{COS}$$ (Eq.7)
where $F_{COSmodel}$ is the modelled COS flux (pmol m$^{-2}$ s$^{-1}$), $\chi_{COS}$ (ppt) and $\chi_{CO2}$ (ppm) are the measured ambient mixing ratios of
COS and $CO_2$ respectively and LRU (-) is the leaf relative uptake rate:
$$LRU = \iota\, e^{\left(\frac{\kappa}{R_{PAR}}\right)}$$ (Eq.8)
where $\iota$ (-) corresponds to the LRU at high light intensity and the parameter $\kappa$ (µmol m$^{-2}$ s$^{-1}$) governs the increase of LRU at
low light conditions. The light dependency of LRU originates from the fact that the COS uptake by the enzyme CA is light-
independent, while the $CO_2$ uptake by Rubisco depends on solar radiation absorbed by leaf chlorophyll (Whelan et al.,
2018;Kooijmans et al., 2019;Wohlfahrt et al., 2012). We determined the parameter $E_0$ by using nighttime data minimizing
the root squared mean error. For the determination of the remaining five unknown model parameters of the two flux
partitioning models we used DREAM, a multi-chain Markov Chain Monte Carlo algorithm (for more detail see Spielmann et
al. (2019)). We calculated the parameters for ~15 day windows but adjusted them to not overlap with a cut of the grassland.
The ecosystem relative uptake (ERU) was calculated using Eq. 1 substituting the GPP with the NEE and using the COS
ecosystem flux for $F_{COS}$.

### 204 2.5.4 Linear perturbation analysis

The relative contribution of the parameters GPP, $F_{COSmodel}$, $\chi_{CO2}$ and $\chi_{COS}$ that drive $\iota$ (Eq. 7) were estimated through a linear
perturbation analysis (Stoy et al., 2006).
The changes in $\iota$ ($\delta\iota$) between the target and the reference window (before the 2$^{nd}$ cut, i.e. 18.06.2015-07.07.2015) are
considered the total derivative of Eq. 7 and can be represented by a multivariate Taylors's expansion where the higher-order
terms are neglected in this first-order analysis:
$$\delta\iota = \frac{\partial\iota}{\partial F_{COSmod}}dF_{COSmod} + \frac{\partial\iota}{\partial\chi_{COS}}d\chi_{COS} + \frac{\partial\iota}{\partial GPP}dGPP + \frac{\partial\iota}{\partial\chi_{CO2}}d\chi_{CO2}$$ (Eq.9)
The relative contributions of the parameters were determined by computing the partial derivatives of Eq. 7.
$$\frac{\partial\iota}{\partial F_{COSmod}} = \frac{\chi_{CO2}}{\chi_{COS}GPP}$$ (Eq.10)
$$\frac{\partial\iota}{\partial\chi_{COS}} = \frac{-\chi_{CO2}F_{COSmod}}{\chi_{COS}^2 GPP}$$ (Eq.11)
$$\frac{\partial\iota}{\partial GPP} = \frac{\chi_{CO2}F_{COSmod}}{\chi_{COS}GPP^2}$$ (Eq.12)
$$\frac{\partial\iota}{\partial\chi_{CO2}} = \frac{F_{COSmod}}{\chi_{COS}GPP}$$ (Eq.13)

### 217 2.6 Ancillary data

Supporting meteorological measurements included $T_{air}$ (RFT-2, UMS, Munich, GER), $T_{soil}$ (TCAV, Campbell Scientific,
Logan, UT, USA), SWC (ML2x, Delta-T Devices, Cambridge, UK), incident solar radiation (CNR-1, Klipp and Zonen,
Delft, NLD), incident photosynthetic active radiation (PAR) (BF2H, Delta-T Devices Ltd, Cambridge, UK) and the
Normalized Difference Vegetation Index (NDVI) sensor (SRS-NDVI, Meter, Pullman, WA, USA). The data were recorded
throughout the whole season as 1 min values and stored as half-hourly means and standard deviations.

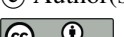
## 3 Results

### 3.1 Environmental conditions

Air temperature ranged between -2 °C and 33 °C with a mean of 13 °C during the study period from 15$^{th}$ of May to first of November (Fig. 1). While the majority of precipitation (total 360.5 mm) fell as rain, we observed an exceptionally late snow event on the 20$^{th}$ of May, which did not melt for almost two days (Fig. 1). Although the VPD reached values of above 2 kPa during 25 days, and plant available water dropped below 38 % on 21 days during the campaign (Fig. 1), we did not observe any relationship with COS (see Fig S1-S2). Due to the removal of the aboveground biomass, the cuts reduced LAI. They also reduced the normalized difference vegetation index (NDVI) (Fig. 1), which further decreased in the subsequent days as a consequence of dying plant parts remaining at the field site (Fig 2 panels a-c). This can also be observed in the webcam photos **(Photo S1-S3)**.

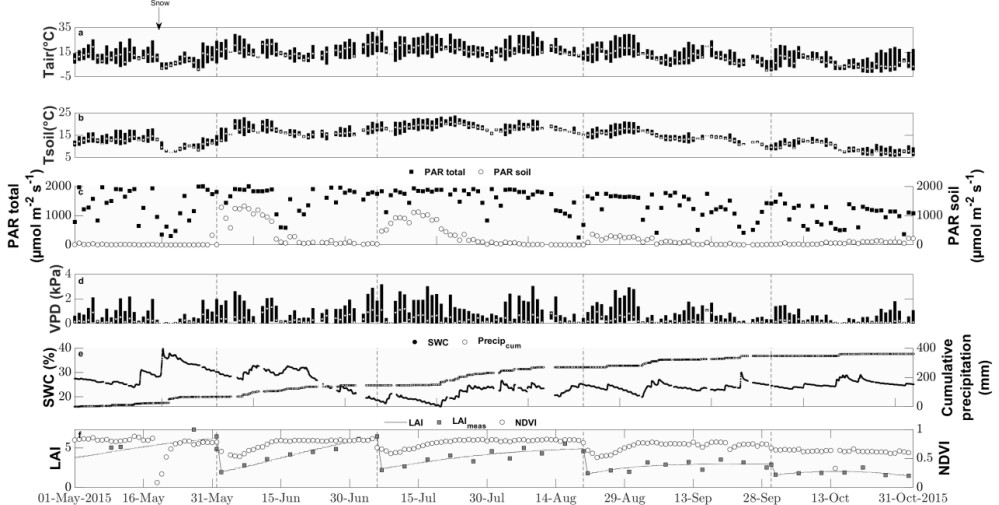

**Figure 1.** Seasonal cycle of ancillary variables. Daily minimum, maximum and median (a) air and (b) soil temperatures (°C) indicated by the lower and upper end of the bars and the white circle, respectively. (c) Daily maximum incident photosynthetic active radiation (µmol m$^{-2}$ s$^{-1}$) reaching the top of the canopy (black squares) and reaching the soil surface (white circles). (d) Daily minimum, maximum and median vapor pressure deficit (kPA) indicated by the lower and upper end of the bars and the white circle, respectively. (e) Soil water content (%) depicted by black squares and cumulative precipitation (mm) depicted by open circles. (f) Modelled leaf area index (black lines), measured LAI (grey squares) and normalized difference vegetation index (open circles).





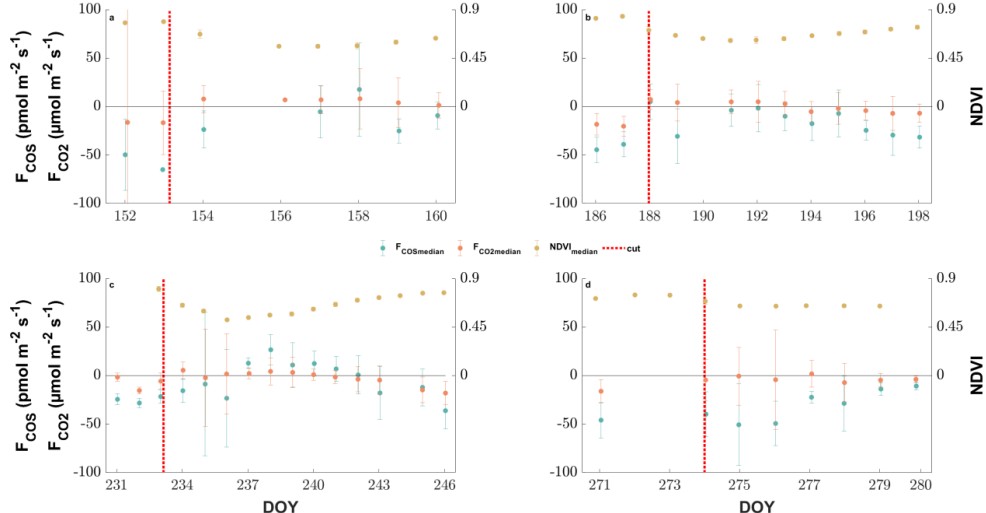

240

**Figure 2:** The response of the daily midday medians of NDVI (yellow circles), COS (blue circles) and $CO_2$ (red circles) ecosystem fluxes around the 4 cutting events (a-d) of the grassland. The errorbars depict the respective median absolute deviations. The cuts are marked by a red dashed line.

244

### 3.2 COS soil flux

The fluxes resulting from the soil chamber measurements ranged from -6.3 to 40.9 pmol $m^{-2}s^{-1}$, with positive fluxes denoting emission (see Fig. 3 panel d).

During nighttime ($R_{SW} = 0$, n = 43), the soils of the grassland acted as a net sink for COS 74.4 % of the time (range of -4.4 to 6.9 pmol $m^{-2}s^{-1}$), whereas soils transitioned to a source in 88.5 % of all daytime measurements ($R_{SW} > 0$, n = 200), reaching the highest fluxes of 40.9 pmol $m^{-2}s^{-1}$ during midday (see Fig. 3 a-c and Fig. S3). This diel pattern was maintained over the course of the season, however with decreasing maximum COS source strength of the soil towards the end of the season (Fig. 3 a-c and Fig. S3). The random forest regression revealed that the most important variable for predicting the soil fluxes was the incident shortwave radiation reaching the soil surface ($R_{SW\text{-}soil}$), accounting for more than 73.53 % of the total variance explained by the final model, while SWC and $T_{soil}$ only accounted for 17.84 % and 8.62 %, respectively. The fast response of the COS soil fluxes to changes in $R_{SW}$ can be seen in Fig. 3 a, where we observed a decrease of $R_{SW\text{-}soil}$ as well as the COS soil flux during a cloudy period, even when the soil temperature still increased. Soil fluxes estimated with the random forest regression ranged from -1.3 to 5.0 pmol $m^{-2}s^{-1}$, reflecting the fact that under real-world conditions very little solar radiation reaches the soil surface. (Fig. 3 e). The resulting emissions peaked during daytime shortly after the cuts when a high proportion of incident radiation was reaching the soil surface, while simulated nighttime fluxes were dominated by uptake (in 93 % of all cases) for the whole season.

261



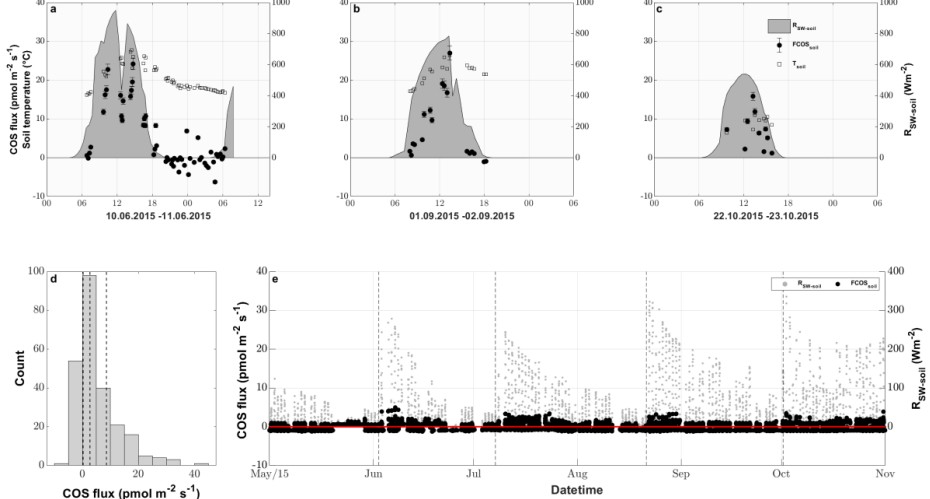

**Figure 3.** COS soil fluxes (pmol m$^{-2}$s$^{-1}$) originating from manual chamber measurements of three selected days (a), (b) and (c) depicted by black circles and open diamonds, respectively, incident shortwave radiation reaching the soil (R$_{SW-soil}$) depicted by the gray area and soil temperature (T$_{soil}$) depicted by empty black bordered squares. (d) Histogram of all conducted COS soil chamber observations with the dashed vertical lines depicting the 25, 50 and 75% quantile. (e) Season plot of the modelled COS soil fluxes (F$_{COSsoil}$) depicted by the black circles, incident shortwave radiation reaching the soil surface (R$_{SW-soil}$) depicted by grey circles and the black dashed lines depicting the cuttings of the grassland.

### 3.3 COS and CO$_2$ ecosystem-scale fluxes

The grassland acted as a net sink for COS during the majority of our study period with 80 % of the COS ecosystem fluxes between -60.2 pmol m$^{-2}$s$^{-1}$ and -12.5 pmol m$^{-2}$s$^{-1}$ during daytime and -41.5 pmol m$^{-2}$s$^{-1}$and -4.6 pmol m$^{-2}$s$^{-1}$ during nighttime. However, we also observed a net release of COS at the field site 4.5 % of the time. The net CO$_2$ fluxes ranged from -20.7 to 3.2 µmol m$^{-2}$s$^{-1}$ and 1.6 to 28.7 µmol m$^{-2}$s$^{-1}$ for 80% of all observation during day and nighttime, with daytime net emissions occurring after the cuttings of the grassland (Fig. 2 a-c and Fig. 4 a). While the COS nighttime fluxes remained unaffected by the cuts (Fig. 4 c), the daytime fluxes showed a high variability (see Fig. 4 b). Especially after the cuts we observed a strong decline in COS uptake and even times where the grassland turned into a net source for COS with midday means of up to 24.5 pmol m$^{-2}$s$^{-1}$ (Fig. 4 b) for up to 8 days after the cut, when the dried litter had already been removed (Fig. 2 a-c). Compared to respiration processes outpacing GPP almost instantaneously after the cuts, the grassland reached its peak COS emission on the day of the cut only in July, whereas the peak was reached five days after the cut in June and August (Fig. 2 a-c). The cut in October led to a reduction in COS uptake, which was lowest three days after the cut (Fig. 2 d). After the fertilization of the field in October the grassland also turned into a source for COS during midday hours for one day (Fig. 4 b). Our flux measurements also included a time when the grassland was covered with snow (on the 20.05.2015), which reduced the COS (and CO$_2$) fluxes to values close to zero. Over the course of the season, we observed a decline in the magnitude of the daytime COS uptake from -50.9 ± 25.0 pmol m$^{-2}$s$^{-1}$ during midday in the first week of May down to -29.6 ± 25.5 pmol m$^{-2}$s$^{-1}$ in the last week of October, which was also correlated with the decline in sink strength and shift to net emission of CO$_2$ from -19.9 ± 8.0 µmol m$^{-2}$s$^{-1}$ to 11.9 ± 36.9 µmol m$^{-2}$s$^{-1}$ (Fig. 4 c). We observed an increase in COS and CO$_2$ fluxes within the growing phases after the cuts only up to an LAI of ~ 4 (-) (Fig. S4-S5), which then levelled out for COS and declined for CO$_2$ due to ecosystem respiration compensating GPP.

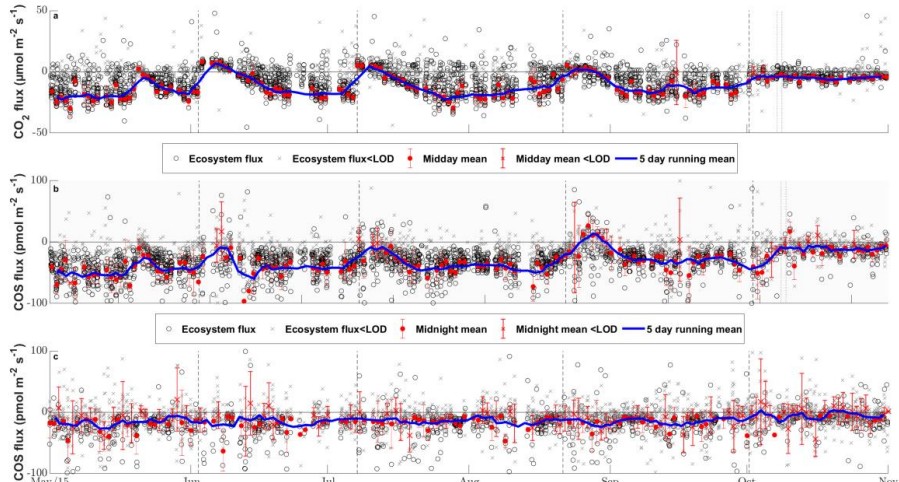

290

**Figure 4:** Seasonal cycle of the half hourly $CO_2$ (a), COS daytime (b) and COS nighttime (c) ecosystem fluxes in µmol $m^{-2}s^{-1}$ and pmol $m^{-2}s^{-1}$ depicted by black circles if they are above the limit of detection (LOD) and grey x's if they are below (Langford et al., 2015). The red circles depict the mean fluxes between 11 a.m. and 2 p.m. CET for (a & b) and between 11 p.m. and 2 a.m. for (c) that are above the LOD, while the red x's indicate means below the LOD. The red error bars depict the ±1standard deviation of the mean. The blue lines depict the running mean (5 days) for the mean fluxes. The black dashed lines depict the cuttings of the grassland.


The seasonal pattern of a decrease in COS sink strength was similar for nighttime fluxes (-18.1± 29.7 pmol $m^{-2}s^{-1}$ to -13.0 ±
22.5 pmol $m^{-2}s^{-1}$) (Fig. 4a). The mean nighttime respiration also decreased over the course of the season from 15.9 ± 28 pmol
$m^{-2}s^{-1}$ to 12.9 ± 31.7 pmol $m^{-2}s^{-1}$ between May and October.
Periods of low (after cuts) and high (before cuts) LAI were compared as diel courses (Fig. 5). Over the course of the day,
both periods were characterized by a mean uptake of COS (Fig 5 c & d). Even though the uptake was similar during
nighttime, the daytime pattern differed considerably. The modelled contribution of the soil to the ecosystem scale COS flux
under high LAI conditions (Fig. 5 d) was minor, contributing between 1 % and 5.5 % of the ecosystem flux during midday
and morning/evening, respectively. In contrast, during low LAI conditions the soil contribution to the ecosystem fluxes
increased during daytime and contributed up to 80.5% of the mean hourly COS ecosystem flux (Fig 5. c). While the
grassland acted as a stronger sink for COS during daytime at a high LAI, reaching peak mean uptake values of up to -41.8
pmol $m^{-2} s^{-1}$± 16.8 pmol $m^{-2} s^{-1}$ during midday, the mean daytime sink strength weakened and we observed close to zero
fluxes during midday in periods of low LAI. The magnitude of the soil flux (2 ± 1 pmol $m^{-2} s^{-1}$) was not high enough to
explain the difference variation of up to -23.7 pmol $m^{-2}s^{-1}$ between the measured COS ecosystem flux and COS flux resulting
from the FP+ model (Fig 5 c), suggesting a missing COS source. For phases of high LAI we saw a good agreement between
modelled and measured COS ecosystem fluxes (Fig 5 d). While the grassland acted as a net sink for $CO_2$ during periods of
high LAI (Fig. 5 b), a combination of a decline in GPP and an increase in RECO turned it into a net source during midday in
periods of low LAI (Fig. 5 a).



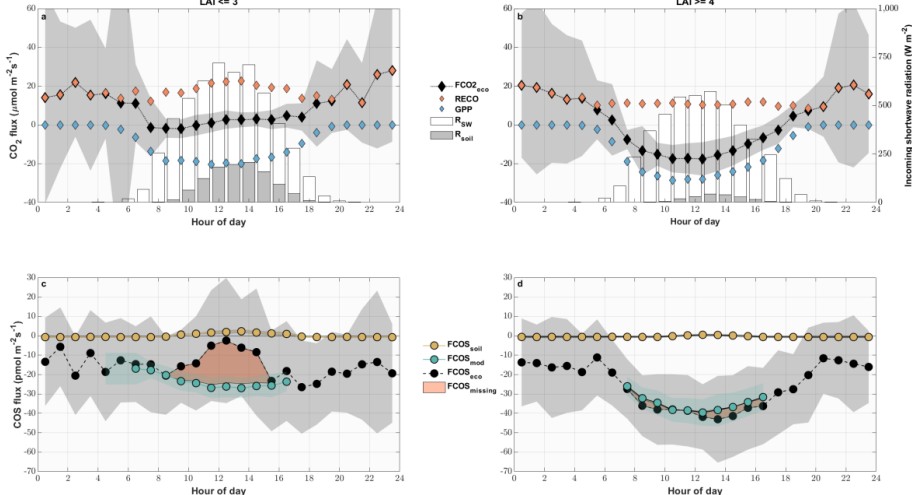


**Figure 5.** Mean diel variation of the measured and modelled $CO_2$ (a & b) and COS (c & d) fluxes for phases of low (LAI <=3) (a & c) and high (LAI >= 4) (b & d). The carats depict the modelled gross primary productivity (blue), the modelled ecosystem respiration (red) and the measured $CO_2$ ecosystem fluxes (black) in µmol m$^{-2}$s$^{-1}$. The circles depict the modelled COS soil flux (yellow), the modelled COS ecosystem flux (turquoise) and the measured $CO_2$ ecosystem fluxes (black) in pmol m$^{-2}$s$^{-1}$. The red area depicts the difference between measured ecosystem flux and the sum of the modelled fluxes. The grey areas depict the ±1 standard deviation of the mean for all the measured fluxes. The white bars depict the diel mean total incoming shortwave radiation (W m$^{-2}$s$^{-1}$) while the grey bars indicate the diel mean shortwave radiation reaching the soil surface.

323

### 3.4 COS mixing ratios above and within the canopy

While the canopy depleted the ambient $\chi_{COS}$ during day as well as nighttime, we found that the $\chi_{COS}$ reached values as low as 134 ppt (depletion of 102 ppt with respect to the mixing ratio at canopy height) during nighttime (see Fig. 6) at the bottom of the canopy in contrast to the midday $\chi_{COS}$, which only went down to 389 ppt (depletion of 125 ppt with respect to the mixing ratio at canopy height). We observed a decrease in $\chi_{CO2}$ (up to 26 ppm) within the most upper layers of the canopy compared to $\chi_{CO2}$ at canopy height during daytime, while $\chi_{CO2}$ increased within the lowest layers compared to $\chi_{CO2}$ at the canopy height due to soil respiration. The above canopy $\chi_{COS}$ increased considerably starting at the onset of the day and reached 587 ppt at 16:00. Over the course of the season the midday ambient $\chi_{COS}$ decreased from 500 ±28 ppt from mid-June to mid-July to 405±29 ppt in October with the trend of increasing $\chi_{COS}$ starting at the end of September (see Fig. S6).


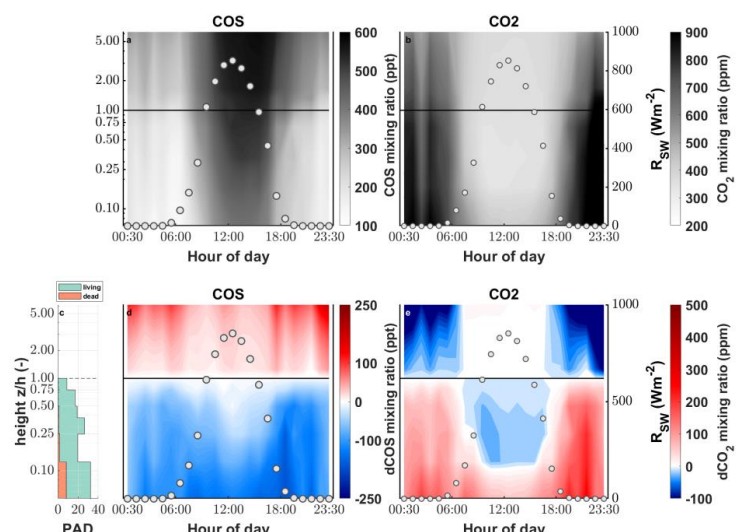

**Figure 6.** Vertical gradient of the (a) COS and (b) $CO_2$ mixing ratio (ppt and ppm, respectively) depicted by the background color between the soil and the eddy covariance tower at 250 cm for one day. The left y axis shows the log of the measurement divided by the canopy height (z/h). The white circles depict the incoming shortwave radiation ($R_{SW}$) in (W m$^{-2}$s$^{-1}$). Plant area density (PAD) split into living (green) and dead (brown) plant material (c). Vertical gradient of the difference between the mixing ratio at canopy height and each measurement height for (d) COS and (e) $CO_2$.

### 3.5 Leaf and ecosystem relative uptake

The LRU at high-light conditions, ι, which we calculated using the FP+ algorithm increased from relatively stable precut levels of 0.9-1.1 (-) after the 2$^{nd}$ and the 4$^{th}$ cut to up to 1.5 (-) (Fig. 7a). After the decrease in ι between the 2$^{nd}$ and the 3$^{rd}$ cut, ι increased steadily until the 4$^{th}$ cut, with the 3$^{rd}$ cut seemingly not having an effect. The reason for the increase in ι after the 2$^{nd}$ and 4$^{th}$ cut was a stronger decrease in GPP than the COS uptake, while both decreased more evenly after the 3$^{rd}$ cut (Fig. 7b). We observed ι in the period before the 4$^{th}$ cut to be influenced not only by a decrease in COS uptake, but also by a decrease in COS mixing ratio (Fig 7b). The mean midday ERUs varied between 1.9 ± 0.1(-) before and 4.6 ± 0.3 (-) after the cuts when excluding and 3.9 ±1.3 (-) when including the first cut. The larger difference between the ERU and ι after the cuts reflect the higher respiration rates of the ecosystem.

Under low light conditions, the LRU increased during pre- and post-cut phases in a similar manner with the last 15-day period in October showing an earlier increase in the morning and evening (see Fig. S7).





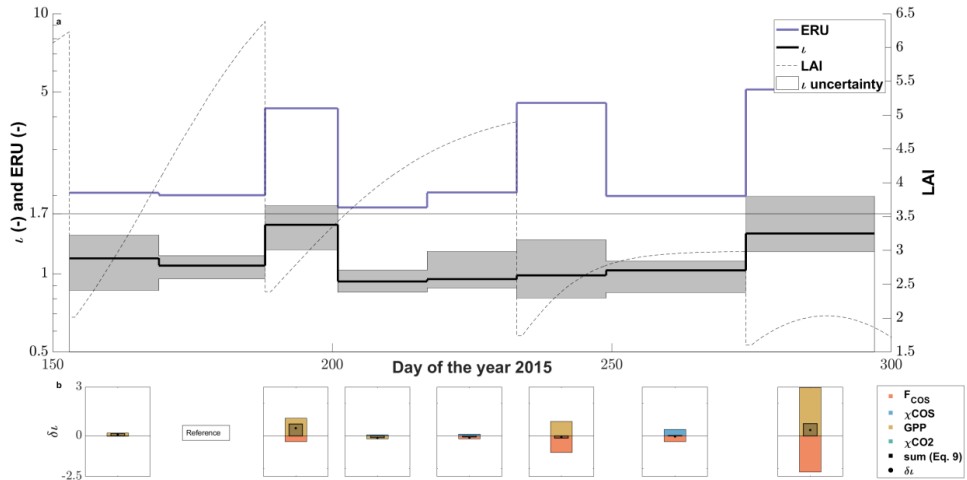

**Figure 7.** (a) The seasonal cycle of ι (black line) with the 95% confidence interval (gray area) resulting from the FP+ model and the midday mean (11 a.m. – 2 p.m. at PAR > 800 µmol m$_{-2}$ s$_{-1}$) ecosystem relative uptake (ERU) (blue line) using the $CO_2$ ecosystem flux for the calculation windows (~15 days adjusted to cuts). The dashed black line depicts the progression of the leaf area index (LAI) of the grassland. (b) The contribution of the drivers ($F_{COS}$, $\chi_{COS}$, GPP and $\chi_{CO2}$) to the changes in ι between all calculation windows and the reference period (DOY 169-188) resulting from the linear perturbation analysis compared to the observed change in ι (δι).

## 4 Discussion

### 4.1 Soil fluxes

The nighttime soil chamber measurements compare well in terms of magnitude with the COS fluxes resulting from studies using dark chambers in agricultural and grassland sites (Whelan et al., 2018;Maseyk et al., 2014;Whelan and Rhew, 2016;Liu et al., 2010) and indicate the soil to be a small sink for COS. The current understanding of COS soil exchange links the COS consumption to soil biota e.g. bacteria and fungi, possessing the ubiquitous enzyme CA (Kesselmeier et al., 1999;Meredith et al., 2019). However, we also found 12 % of all nighttime fluxes to be emission. The origin of COS in soils is still highly debated, but comparisons of untreated and sterilized soils suggest yet unknown abiotic processes (Meredith et al., 2019;Kitz et al., 2019).

During daytime, the soil inside the chambers emitted COS at rates of up to 40.9 pmol m$^{-2}$s$^{-1}$. These rates lie at the upper end of recently stated values of agricultural and grassland sites (Whelan et al., 2018;Kitz et al., 2017;Maseyk et al., 2014;Liu et al., 2010). Partly, this can be attributed to the type of chambers we used and their deployment. We allowed the full spectrum of incoming radiation to reach the soils surface, whereas most other studies used dark chambers. Therefore we were able to capture the influence of COS emission processes coupled to thermo- and photo production on our COS soil fluxes (Whelan and Rhew, 2015;Kitz et al., 2019;Meredith et al., 2018). This also led to lower peak soil emissions of COS at the end of the season, when the incoming radiation declined. Our modelled COS soil fluxes peak at about 12% of the maximum emissions retrieved from the soil chambers. This is owed to the difference in incident radiation reaching the soil surface between the fluxes resulting from chamber measurements and our model. For the chambers, the aboveground biomass was removed, whereas our modelled fluxes were adjusted for undisturbed canopy conditions. In the gradient mixing ratio data, during pre-cut conditions, we also did not see an increase in COS mixing ratio within the canopy, which would have been a hint for a soil COS source.



Another factor contributing to the high COS soil emissions might be the yearly fertilization using slurry, as high nitrogen
content in soils has been linked to a higher source strength of COS (Kaisermann et al., 2018). This agrees well with the study
of Kitz et al. (2019), who found a correlation between increased soil nitrogen content and soil COS emission in a laboratory
experiment with samples taken from the grassland at two different dates (i.e. June and September).

### 4.2 Ecosystem fluxes

Our observations show that the agriculturally used grassland acted as a major sink for COS during the growing season. The
fluxes fit well within or even exceeded the COS uptake rates of published grassland and agricultural sites during their
growing phases (Billesbach et al., 2014;Whelan and Rhew, 2016;Geng and Mu, 2004). The late snow event that occurred in
the peak growing season almost completely inhibited the exchange of $CO_2$ and COS, as the snow acted as a diffusion barrier
for these compounds (Björkman et al., 2010).
The cuttings and the consecutive drying of the above ground plant material at the site had a major influence on the COS
exchange. During these events the grassland turned into a source for $CO_2$ and COS. This has also been reported at
agricultural fields in phases of senescence (Maseyk et al., 2014;Billesbach et al., 2014). Although the soil was a strong
source for COS, caused by the high $R_{soil}$ and $T_{soil}$ (Whelan and Rhew, 2015;Kitz et al., 2019;Meredith et al., 2018), and the
sink strength of the grassland was low due to the reduced aboveground biomass, soil fluxes did not explain the emission on
ecosystem level (see Fig. 5a). As plants contain precursors involved in COS emission processes, e.g. methionine and
cysteine (Meredith et al., 2018), the plant litter and dying plant parts remaining at the site after the cuts might be the missing
source of COS. Laboratory tests of the soil of the grassland have shown that a mixing of dried litter and soil lead to a strong
but short-lived emission peak of COS (Kitz et al., 2019). Alternatively, the cutting of the grassland might induce stress
mediated COS production in the remaining living plant parts (Bloem et al., 2012;Gimeno et al., 2017). The delay in the peak
COS emissions at ecosystem scale after the cuts could indicate that some yet unknown biotic or abiotic processes take
several days to release COS.
We also observed another COS emission event shortly after the fertilization of the grassland towards the end of the growing
season. The increase of available nitrogen (Kaisermann et al., 2018) and COS precursors introduced to the soil in the form of
cattle slurry (Hörtnagl et al., 2018) might have triggered the COS emission by biotic or abiotic processes.
Due to the independence of CA to catalyze COS without $R_{PAR}$ (Stimler et al., 2011), the grassland remained a sink for COS
during nighttime. Again, the soil sink was too small to explain the total COS exchange (Fig. 5), which indicates that the plant
stomata were not fully closed (Kooijmans et al., 2017) and were responsible for the majority of the COS uptake. The
minimum or residual stomatal conductances at the field site in Neustift have been reported to be between 10 and 65 mmol m$^-$
$^2$ s$^{-1}$ depending on the species (Wohlfahrt, 2004).
Although we observed phases of high VPD and low SWC (Fig. 1), they did not lead to a decrease in $CO_2$ and COS
ecosystem fluxes (Fig. S1-S2), which has already been observed for the grasslands $CO_2$ and $H_2O$ fluxes between 2001 and
2009. The species located at the site were insensitive to progressive drought conditions (Brilli et al., 2011).

### 4.3 COS mixing ratios

The continuous decrease in above-canopy $\chi_{COS}$ from ~500 ppt (in May) to ~400 ppt (in October) is within the range of
published records observing mixing ratios to decrease from 465 (in summer) to 375 ppt (in winter) (Kuhn et al., 1999). This
pattern is typical for the northern hemisphere and the COS drawdown by terrestrial ecosystems (Montzka et al., 2007). We
found the lowest $\chi_{COS}$ at the end of September, which coincides with the lowest ambient mixing ratios of COS, measured in
Ireland, the closest COS observation site Mace Head (MHD) of NOAA, on the 6[th] of October.
Gradient observations of the diurnal cycle revealed a continuous decrease of $\chi_{COS}$ from the atmosphere (> 500ppt) down to
the soil reaching very low concentrations of 134 ppt during nighttime. Low values like this have also been reported by





Rastogi et al. (2018), who measured a mean $\chi_{COS}$ minimum of 152 ppt at 1 m above the soil within an old growth forest. The
difference in concentrations during day and nighttime originates from changes in the height of the planetary boundary layer
(PBL). While the PBL is shallow during nighttime and the COS mixing ratio decreases due to sink strength of the grassland,
at the onset of the day, the PBL layer height increases fast and COS rich air is transported down to the ecosystem. Even
though $CO_2$ and COS share a similar pathway into plants, reflected by their respective decrease in the mixing ratios within
the canopy, we saw a difference at the lower levels of our gradient analysis. We only observed an increase in $CO_2$ mixing
ratios, caused by the release of $CO_2$ through respiration processes in the soil, whereas COS mixing ratios further declined
down to the soil surface. This supports our soil model, which predicted only minor COS fluxes under conditions of high
LAI, when only a small portion of incident radiation was hitting the soil surface.
**4.4 LRU**
The parameter ι varied between 0.9 (0.8-1.0) (-) and 1.5 (1.2-1.8) (-) during the campaign, where cuts of the grassland tended
to result in higher values and places this study at the lower end of a recent compilation of all published leaf-level LRUs, that
put 95% of all data between 0.7 (-) and 6.2 (-) with a median of 1.7 (-) (Whelan et al., 2018) and also lower than the LRU of
2.53 (-) estimated for grasslands by Seibt et al. (2010). The seasonal trend of the LRUs was strongly influenced by the
cutting of the grass and can be attributed mainly to changes in the ratio of COS uptake to GPP. However, we also observed a
strong decline in the ambient mixing ratio of COS, which also had an equally strong influence on the change in ι as the COS
flux for the 15 day window before the last cut (Fig 7 b).
Even though the changes in ι can be explained, it is important to keep in mind that the grassland was a source for COS on
ecosystem level after the cuts. For the calculation of LRUs we had to remove those observations from the data since they
would yield negative values (see Eq.8). This indicates that the unknown source strength after cuts likely decreases the post-
cut ι's.
**5 Conclusion**
Due to the management interventions at the grassland site, the leaf area development was decoupled from seasonal changes
in environmental forcing. This allowed us to measure concurrent $CO_2$ and COS fluxes at soil and ecosystem level for
multiple growing periods within one season. The LAI on seasonal scale as well as incoming solar radiation on hourly to
seasonal scales determined whether soils were a source or a sink for COS. The incoming shortwave radiation reaching the
soil surface had a decisive influence on the COS soil surface flux and thus supports our hypothesis H4. The covariance
between the daytime $CO_2$ and COS fluxes on daily to seasonal level was high and the fluxes only diverged after the cuts,
leading to higher LRUs. Beside the perturbations of the ecosystem, the sink strength of the grassland was high for COS and
declined over the course of the season (H1). The COS emissions at ecosystem scale shortly after the cuts, which could not be
explained by the soil source, raise questions about other unknown mechanisms of COS production within ecosystems (H2).
With the exception of short periods after the cuts, the LRUs under high light conditions were relatively constant during the
season, indicating a good correlation between the COS flux and GPP under stable conditions (H3).
**6. Data availability**
Data and materials availability: Will be uploaded to https://zenodo.org/.



## 7. Author contributions

Felix M. Spielmann: Data curation, Formal analysis, Investigation, Methodology, Software, Visualization, Writing – original draft

Albin Hammerle: Data curation, Investigation, Software, Writing – original draft

Florian Kitz: Data curation, Formal analysis, Investigation, Methodology, Software, Writing – original draft

Katharina Gerdel: Investigation, Software, Writing – original draft

Georg Wohlfahrt: Conceptualization, Funding acquisition, Investigation, Methodology, Project administration, Software, Supervision, Writing – original draft

## 8. Competing interests

The authors declare no competing financial interests.

## 9. Acknowledgements

This study was financially supported by the Austrian National Science Fund (FWF; contracts P26931, P27176, and I03859), the Tyrolean Science Fund (contract UNI-0404/1801), and the University of Innsbruck (Infrastructure funding by Research Area Alpine Space-Man and Environment to G. W). Financial support to F. M. S. was provided through a PhD scholarship by the University of Innsbruck. We thank family Hofer (Neustift, Austria) for kindly granting us access to the study site. COS flask data were provided by the Global Monitoring Division of the Nationals Oceanic and Atmospheric Administration's Earth System Research Laboratory (NOAA ESRL/GMD). The authors declare no competing financial interests.

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
