# Peer review of "Seasonal dynamics of the COS and CO2 exchange of a managed"

_Biogeosciences, 2020_

## Referee Comment (RC1) · Anonymous Referee #1 · 3 Apr 2020

This manuscript conducted concurrent ecosystem scale CO2 and COS flux measurements above an agriculturally managed temperate mountain grassland to test the applicability of COS as a tracer for GPP at larger temporal scales. The results indicated that a high correlation between the COS flux and GPP across the growing season under high light conditions with rather stable LRU except for the short periods after the cuts. Additionally, a new finding was also present in the manuscript, e.g., the grassland turned into a net source for CO2 and COS on ecosystem level after the cuts especially during daytime under higher incident radiation hitting the soil surface. This reviewer recommends the manuscript be published in the journal after considering the following minor comments:

Line 31. Are you sure that "the summer drawdown for COS is 6 times stronger than for

[Figure]

CO2"? The magnitude of the times seems too large to be believable. Line 130. "while air was sucked through the chamber to the QCL at a flow rate of 1.5 l min-1". The heights of air inlets for the chamber and ambient environment should be noted because remarkable vertical distribution of COS mixing ratio near the ground was observed in this study. If the height of air inlet for the chamber was within the canopy of the grass, the COS uptake flux would be largely overestimated, e.g., the COS mixing ratio could drop to 134ppt within the canopy in comparison with about 500ppt over the canopy. Line 228. What's the plant available water? Fig. 1 only presents the SWC (%) which is below 38% during almost all days, rather than 21 days. Line 248. "During nighttime (RSW = 0, n = 43), the soils of the grassland acted as a net sink for COS 74.4 % of the time" is better replaced by "During nighttime (RSW = 0, n = 43), 74.4 % of the COS emission fluxes were negative, implying soils of the grassland acted as a net sink for COS". Line 263. Why did you use both circles and open diamonds for depicting COS soil fluxes? What's the difference between them? Lines 276-278. "Especially after the cuts we observed a strong decline in COS uptake and even times where the grassland turned into a net source for COS with midday means of up to 24.5 pmol m-2s-1 (Fig. 4 b) for up to 8 days after the cut, when the dried litter had already been removed (Fig. 2 a-c)". This sentence is suggested to be replaced by "Especially after the cuts we observed a strong decline in COS uptake ((Fig. 4 b)) and the grassland even turned into a net source for COS in middays (Fig. 2 a-c) with a highest emission flux of 24.5 pmol m-2s-1 in August after the cut.". Lines 280-281. "The cut in October led to a reduction in COS uptake, which was lowest three days after the cut (Fig. 2 d)". The description seems to be inconsistent with the Fig. 2d. Lines 297-298. I don't understand the meaning of the sentence. Fig. 4a is the seasonal cycle of CO2, rather than COS. Lines 325-328. I wonder why the COS mixing ratio dropped so large during the nighttime when the COS uptake was much less than that during midday. Lines 375-377. I don't understand the logic of this sentence. Because the chamber enclosed both soil and the residual grass after the cuts, the COS emission under sunlight irradiation might be due to the residual rather than the soil itself, e.g., the photochemical formation

of COS from the possible liquid released from the cut grasses (JGR, 109, D13301, doi:10.1029/2003JD004206, 2004; JES, 5 1 ( 2 0 1 7 ) 1 4 6 – 1 5 6).  If the COS emission was ascribed to soil, the authors are suggested to verify it by using a flow tube method under dark and irradiation conditions. Line 413. Why did the lowest COS mixing ratio appear in winter when vegetation COS uptake is relatively low?  Lines 419-421: The above sentences didn't mention the difference in concentrations during day and nighttime. Lines 421-422. Considering the much stronger COS uptake by the grass in daytime than in nighttime, COS mixing ratio above the canopy should decrease in daytime, rather than nighttime despite of the variation of PBL.

---

## Referee Comment (RC2) · Anonymous Referee #2 · 14 Apr 2020

Spielmann et al. presented COS and CO2 flux and concentration measurements of a season long campaign on ecosystem, soil and canopy levels in a managed grassland in Neustift, Austria. The results are of particular interesting due to the four times of cuttings of the grass, providing opportunities to study the disturbed grassland. The collected dataset is rather comprehensive, followed by thoughtful analyses. The paper is well structured and clearly written, and is suitable for the journal of Biogeosciences. The reviewer suggests publication after addressing the following comments.

General comments:

1. Definition of "LRU on ecosystem scale": note that most LRUs in the literature were derived from branch chamber measurements, and were then used in the relationship between Fcos and Fco2 (Eq.1), with the implication/assumption that LRUs derived

from branch chamber measurements are representative of the entire canopy. Here the authors infer the LRU (of the entire canopy) from ecosystem flux measurements. Please clarify this.

2. $CO_2$ observations: IRGA $CO_2$ measurements were used in the analyses. I believe that the QCL also measured $CO_2$. Were those data used somehow? If IRGA $CO_2$ measurements were calibrated to the WMO scale, $CO_2$ should be reported as mole fractions instead of mixing ratios, because the WMO scale (NOAA calibration gases) is reported on mole fractions. The difference between mole fractions and mixing ratios is significant for $CO_2$, and not significant for COS.

3. What are the reasons for the relatively low enhancements of daily maximum PAR values reaching the soil surface after the third and the fourth cuts (Figure 1)? These are not consistent with the "incident shortwave radiation reaching the soil surface" in Figure 3e.

4. Fcosmedian turned to positive after the third cutting while remained largely negative after the fourth cutting (Figure 2c&d), given that COS soil fluxes would be both positive. What could explain the difference here?

5. High-light conditions: what is the definition of high-light conditions? How sensitive is the estimated LRU at high light intensity to the choice of high-light conditions?

Other technical comments:

Line 111: I think it is more likely by a GC-MS than a GC, please double check.

L154: The unit of RSW-soil should be Wm-2, and for other places as well.

L165: obtain-high resolution → obtain high-resolution

L191: Eq.7 was developed in earlier studies, please refer to the original work.

L198-203: It will read better if these are moved to after L188.

[Figure]

L230: It needs a bit more explanation of NDVI, what does it indicate?

Figure 3 caption. open diamonds?

L312: why is an increase in RECO expected?

L319: should be COS instead of CO2

L433-435: LRU is a normalized ratio, and should not depend on the ambient COS. I do not get the point here.

L437-439: Please specify which are the exact "those observations". Figure 4 indicates that low COS fluxes took place shortly after the cuttings, which coincides with COS emissions from soils after the cuttings.

L419-422: It may be worth pointing out that the vertical gradient of COS between the canopy level and below the canopy levels exists throughout the day and night, but that of CO2 does not.

---

## Referee Comment (RC3) · Anonymous Referee #3 · 15 Apr 2020

This is a nice paper describing COS and CO2 fluxes in an alpine managed grassland. The authors also evaluate their model that includes a light dependent COS/CO2 relative uptake to account for differences in the uptake pathways. They also observe a soil emission of COS after harvest and discuss photolytic sources of that OCS emission.

I recommend publication after the authors address what I hope are minor comments described below.

Minor comments in general: There seems to be a really strong gradient within the grass canopy. Would the really low COS above the soils (100-200 ppt) influence the COS flux? Out of interest, what does the FCOS/[COS] (COS deposition velocity) look like? I also think the concentration discussion (Sections 3.4, Fig 6, 4.3) should come before the flux discussion. It really sets the context to fully appreciate the flux discussion.

Data needs to be made public before publication! Make sure in the final version that the text in the figures is big enough. I was having to zoom in a lot to read things.

I'm really impressed at how well the FP+ model works for grass (Fig 5b/d). What drives the large change in CO2 variability between day and night? Has the data been filtered for u*? Has any of this large variability been taken into account in the Reco vs temp calculation for GPP uncertainty (something to think about in future if not?).

There is a little repetition with the Results and Discussion being separate. I wouldn't object if the authors decided to combine both and tightened the text up. But obviously that's just a suggestion.

Minor comments by line number: 14: soil flux 31: do you mean relative uptake? COS is in ppt vs CO2 in ppm 38: Extra bracket 86: What kind of fertilizer (dairy? beef? pig?)? And when was it fertilized previously? Before the winter? 140: Ambient COS from what height? There is a massive COS gradient so this will be important. 160: I think this needs more explanation. What does an OBB represent? Is that good? Not good? If you aren't going into enough detail for readers to evaluate the model, then cut it. It's kind of hanging there with not enough info. And most of the packages mentioned will represent some mathematical approach to data analysis. Since packages come and go, it would be really helpful to have a sentence or two about what these packages actually represent. 168: What heights along the tower were the gradients sampled from? How often were they sampled vs eddy flux sampling? 173: Was the eddy flux data filtered for insufficient turbulence? If so, what u* filter was applied? How was the u* threshold quantified? A plot of the FCOS and FCO2 vs u* would be helpful here to understand the micro met dynamics for the site. 329: What does the [CO2] drop down to? Is there a relationship between u*/turbulence and the d[COS] and d[CO2]? That would be an interesting figure to see. 422: How long does the morning increase in COS last for? Do you start to see a decrease in COS as the daytime uptake influences the air in the valley? Other sites have also seen this morning peak in COS. Maybe include a reference to those here. (e.g. Redwoods, Harvard Forest, etc).

---

## Author Comment (AC1) · 5 May 2020

**Line 31. Are you sure that "the summer drawdown for COS is 6 times stronger than for CO2"? The magnitude of the times seems too large to be believable.**

*The exact wording of the cited paper Montzka et al. (2007) is:*

*'However, while reduced mixing ratios of $CO_2$ during the NH growing season represent a balance between vegetative uptake and total respiration (i.e. NEP), the percentage reduction on COS mixing ratio is 4-6 times (5.5+/- 1.6) larger during June- August (calculated relative to mixing ratios measured at 4-8 km asl) (Figure 6c).'*

*We will reword the sentence to more accurately correspond to the cited paper (Montzka 2007).*

**Line 130. "while air was sucked through the chamber to the QCL at a flow rate of 1.5 l min-1". The heights of air inlets for the chamber and ambient environment should be noted because remarkable vertical distribution of COS mixing ratio near the ground was observed in this study. If the height of air inlet for the chamber was within the canopy of the grass, the COS uptake flux would be largely overestimated, e.g., the COS mixing ratio could drop to 134ppt within the canopy in comparison with about 500ppt over the canopy.**

*The intake height was at 0.12 m above the ground and thus within the canopy. The COS concentration inside the chamber was thus similar to what the undisturbed soil would experience, which avoids fluxes being biased high when COS-enriched air from above the canopy would be used. We will include this information in the method section.*
*However, we also have to disagree with the comment on the overestimation of the uptake flux if the intake was within the canopy. If there already is a lower ambient mixing ratio, we would expect a decrease in uptake, as the gradient of the COS mixing ratio would be smaller. To take measurements closer to undisturbed conditions, the intake height should probably be within the canopy.*

**Line 228. What's the plant available water? Fig. 1 only presents the SWC (%) which is below 38% during almost all days, rather than 21 days.**

*The SWC in Fig.1 will be replaced with the plant available water, which falls below 50 % during 111 days.*

[Figure]

**Line 248.** "During nighttime (RSW = 0, n = 43), the soils of the grassland acted as a net sink for COS 74.4 % of the time" is better replaced by "During nighttime (RSW = 0, n = 43), 74.4 % of the COS emission fluxes were negative, implying soils of the grassland acted as a net sink for COS".

*The sentence will be changed as suggested.*

**Line 263.** Why did you use both circles and open diamonds for depicting COS soil fluxes? What's the difference between them?

*We will remove the depiction about the open diamonds, which are not present in the plots.*

**Lines 276-278.** "Especially after the cuts we observed a strong decline in COS uptake and even times where the grassland turned into a net source for COS with midday means of up to 24.5 pmol m-2s-1 (Fig. 4 b) for up to 8 days after the cut, when the dried litter had already been removed (Fig. 2 a-c)". This sentence is suggested to be replaced by "Especially after the cuts we observed a strong decline in COS uptake ((Fig. 4 b)) and the grassland even turned into a net source for COS in middays (Fig. 2 a-c) with a highest emission flux of 24.5 pmol m-2s-1 in August after the cut.".

*We will replace the sentence according to your suggestion. To keep the crucial information about the grassland turning into a net source for up to 8 days after the cut, we will add an additional sentence to the manuscript.*

**Lines 280-281.** "The cut in October led to a reduction in COS uptake, which was lowest three days after the cut (Fig. 2d)". The description seems to be inconsistent with the Fig. 2d.

*We agree that the lowest COS uptake did occur later than 3 days after the cut. However, this is also related to the overall decline in COS uptake by the grassland at the end of the season. We see no recovery*

*of the COS flux after the last cut. We will rephrase this in the manuscript and include more data points to Fig. 2 d):*

*The cut in October led to a reduction in COS uptake, which declined across several days and did not recover, as the end of the season was reached (Fig. 2 d & Fig. 4 b).*

[Figure]

**Lines 297-298. I don't understand the meaning of the sentence. Fig. 4a is the seasonal cycle of CO2, rather than COS.**

*We will change Fig 4a to 4c and add Fig 4a to subsequent sentence dealing with the seasonal response of respiration.*

**Lines 325-328. I wonder why the COS mixing ratio dropped so large during the nighttime when the COS uptake was much less than that during midday.**

*Compared to the constant influx of COS rich air during daytime, due to the increased boundary layer (see line 422), this influx stops during nighttime and COS gets depleted within the canopy, even when the COS uptake of the ecosystem is lower than during daytime. The strong input of COS rich air during daytime has also been reported by other studies (e.g. Rastogi 2018 – Ecosystem fluxes of carbonyl sulfide in an old-growth forest: temporal dynamics and responses to diffuse radiation and heat waves). We will add the reference to the manuscript.*

**Lines 375- 377. I don't understand the logic of this sentence. Because the chamber enclosed both soil and the residual grass after the cuts, the COS emission under sunlight irradiation might be due to the residual rather than the soil itself, e.g., the photochemical formation of COS from the possible liquid released from the cut grasses (JGR, 109, D13301, doi:10.1029/2003JD004206, 2004; JES, 5 1 ( 2 0 1 7 ) 1**

**4 6 – 1 5 6). If the COS emission was ascribed to soil, the authors are suggested to verify it by using a flow tube method under dark and irradiation conditions.**

This sentence will be removed.

**Line 413. Why did the lowest COS mixing ratio appear in winter when vegetation COS uptake is relatively low?**

*During winter, no strong emission fluxes are expected to originate from vegetation and soils. The mixing ratios rather depend on the transport of COS enriched air from oceans, which are also highest in summer (see Montzka 2007).*

**Lines 419-421: The above sentences didn't mention the difference in concentrations during day and nighttime.**

*We will add the sentence:*

*Even though the COS mixing ratio at the layer closest to the soil were higher during day than during nighttime, the absolute decrease in COS was lower during nighttime due to partial stomatal closure (see Kooijmans 2017 – Canopy uptake dominates nighttime carbonyl sulfide fluxes in a boreal forest). The absolute difference in concentrations during day and nighttime originate from changes in the height of the planetary boundary layer (PBL).*

**Lines 421-422. Considering the much stronger COS uptake by the grass in daytime than in nighttime, COS mixing ratio above the canopy should decrease in daytime, rather than nighttime despite of the variation of PBL**

*Several studies (e.g. Rastogi 2018 – Ecosystem fluxes of carbonyl sulfide in an old-growth forest: temporal dynamics and responses to diffuse radiation and heat waves) showed that the PBL is the main influence on sub-diurnal variability in COS mixing ratio. The incomplete stomatal closure as well as the soil sink cause the nighttime decrease in mixing ratio as there is no influx of COS rich air from the atmosphere. The stronger daytime drawdown can also be observed in the gradient analysis as the decrease in COS mixing ratio, from to the canopy height down to the soil was higher during daytime (125 ppt) compared to the nighttime decrease (102 ppt).*

*This information is already present in the manuscript; see line 325-328 and 419-423.*

---

## Author Comment (AC2) · 5 May 2020

**1. Definition of "LRU on ecosystem scale": note that most LRUs in the literature were derived from branch chamber measurements, and were then used in the relationship between Fcos and Fco2 (Eq.1), with the implication/assumption that LRUs derived from branch chamber measurements are representative of the entire canopy. Here the authors infer the LRU (of the entire canopy) from ecosystem flux measurements. Please clarify this.**

*We will add that the LRU is calculated using eddy fluxes without the need to use chambers to the introduction.*

**2. CO2 observations: IRGA CO2 measurements were used in the analyses. I believe that the QCL also measured CO2. Were those data used somehow? If IRGA CO2 measurements were calibrated to the WMO scale, CO2 should be reported as mole fractions instead of mixing ratios, because the WMO scale (NOAA calibration gases) is reported on mole fractions. The difference between mole fractions and mixing ratios is significant for CO2, and not significant for COS.**

*The COS and $CO_2$ fluxes were calculated using the QCL data as stated in section 2.5.2. We followed the processing steps of Gerdel et al. 2017 to retrieve the fluxes using the same filters, which as stated by Gerdel et al. 2017 has the advantage that the influence of the high pass filter on the ecosystem relative uptake (ERU) largely cancels out, if applied on COS as well as $CO_2$. The ambient COS and $CO_2$ concentrations both originated from the QCL data, which puts out mixing ratios. We will change the method section accordingly since neither $CO_2$ nor $H_2O$ fluxes of the IRGA were used in the final version of the manuscript. We apologize for the confusion.*

**3. What are the reasons for the relatively low enhancements of daily maximum PAR values reaching the soil surface after the third and the fourth cuts (Figure 1)? These are not consistent with the "incident shortwave radiation reaching the soil surface" in Figure 3e.**

*The data of the PAR reaching the soil surface in Fig 1 originated from a PAR sensor that was likely overgrown by short vascular plants and mosses growing directly at the soil surface at the end of the season. We will change the data from this sensor to the data of Fig 3e, which was calculated using the Beer-Lambert law (see line 151).*

**4. Fcosmedian turned to positive after the third cutting while remained largely negative after the fourth cutting (Figure 2c&d), given that COS soil fluxes would be both positive. What could explain the difference here?**

*The modelled soil fluxes were always relatively small compared to the ecosystem scale fluxes and shouldn't be the reason for the difference between fig 2c&d. Also, there is less incoming solar energy at the end of the season, likely also decreasing the emission strength of the residual litter. It might also be connected to a decrease in soil nitrogen content over the course of the season, as the grassland is only fertilized at the end of the growing season.*

*We will add a sentence containing this to the discussion.*

**5. High-light conditions: what is the definition of high-light conditions? How sensitive is the estimated LRU at high light intensity to the choice of high-light conditions?**

The parameter "iota" – LRU under high light conditions results from equation 8. The second parameter "kappa" controls the exponential decrease of LRU when the incoming photosynthetic active radiation (PAR) is decreasing and limiting GPP but not the COS flux.

$$LRU = \iota\, e^{(\frac{\kappa}{R_{PAR}})}$$

While mathematically iota is only obtained at infinitely high PAR, in practice above about 800 PAR only insignificant change in the ecosystem relative uptake, reflecting the relationship between the COS and the $CO_2$ flux, can be observed (see attached figure).

[Figure]

We will include the definition for high light into the methods part.

**Other technical comments:**

**Line 111: I think it is more likely by a GC-MS than a GC, please double check.**

We will change GC to GC-MS within the revised document.

**L154: The unit of RSW-soil should be Wm-2, and for other places as well.**

We will change this according to the reviewer comment.

**L165: obtain-high resolution ! obtain high-resolution**

We will change this according to the reviewer comment.

**L191: Eq.7 was developed in earlier studies, please refer to the original work.**

We will change this according to the reviewer comment and add (Sandoval-Soto 2005) as reference.

**L198-203: It will read better if these are moved to after L188.**

*We will change this according to the reviewer comment.*

**L230: It needs a bit more explanation of NDVI, what does it indicate?**

*We will change the manuscript accordingly:*

*They also reduced the normalized difference vegetation index (NDVI) (Fig. 1), which is a measure of canopy greenness (Tucker,1979).*

**Figure 3 caption. open diamonds?**

*We will remove the text part about the open diamonds, which are not present in the figure.*

**L312: why is an increase in RECO expected?**

*Even though there is a reduction in plant respiration, the increase in incoming radiation reaching the soil surface leads to an increase in soil temperature and consequently soil respiration (see Fig.5a). We will add this information to the manuscript.*

**L319: should be COS instead of CO2**

*We will change this according to the reviewer comment.*

**L433-435: LRU is a normalized ratio, and should not depend on the ambient COS. I do not get the point here.**

*This is not quite right. LRU is calculated in order to normalize for differences in COS (and $CO_2$) concentrations, which affect the fluxes. For the same COS and CO2 flux and the same CO2 concentration, LRU will differ whether the ambient COS concentration is 400 or 500 ppt. This is what we quantify in the linear perturbation analysis and what this sentence refers to.*

**L437-439: Please specify which are the exact "those observations". Figure 4 indicates that low COS fluxes took place shortly after the cuttings, which coincides with COS emissions from soils after the cuttings.**

*We will clarify this by changing the sentence to:*

*For the calculation of LRUs we had to remove the canopy flux data containing COS and/or $CO_2$ emissions since they would yield negative values for LRU (see Eq.8).*

**L419-422: It may be worth pointing out that the vertical gradient of COS between the canopy level and below the canopy levels exists throughout the day and night, but that of CO2 does not.**

*We will add the sentence:*

*Compared to the consistent decrease of COS below the canopy level during day and nighttime, the gradient for $CO_2$ reverses during nighttime due to ongoing respiration processes while plants are not photosynthetically active.*

---

## Author Comment (AC3) · 5 May 2020

**Minor comments in general:**

**There seems to be a really strong gradient within the grass canopy. Would the really low COS above the soils (100-200 ppt) influence the COS flux?**

*Yes, since the exchange across the soil surface is driven by the concentration gradient between the ambient air just above the soil surface and within the soil. We will add a sentence containing this information to the discussion.*

**Out of interest, what does the FCOS/[COS] (COS deposition velocity) look like?**

*We attach a plot of the COS deposition velocity. We will also provide this in the revised supplement.*

[Figure]

**I also think the concentration discussion (Sections 3.4, Fig 6, 4.3) should come before the flux discussion. It really sets the context to fully appreciate the flux discussion.**

*We agree and we will move the parts accordingly.*

**Data needs to be made public before publication! Make sure in the final version that the text in the figures is big enough. I was having to zoom in a lot to read things.**

*The data is online now and the font size of the text within the figures will be increased.*

**I'm really impressed at how well the FP+ model works for grass (Fig 5b/d).**

*Thank you, we were also very happy with the fluxes resulting from the model.*

**What drives the large change in CO2 variability between day and night?**

*As shown by Wohlfahrt et al. (2005), the large variability of NEE during nighttime conditions is due to the combination of low wind speeds and stable stratification which results in highly intermittent CO2 fluxes compared to daytime. On a half-hourly basis, fluxes may even be negative (i.e. net uptake of CO2), which is biologically impossible, but results from the intermittent nature of the CO2 exchange and is typically compensated for by large emission fluxes in a subsequent averaging period. As recommended by Wohlfahrt et al. (2005), CO2 fluxes were filtered for u\*, but not for the sign of the fluxes in order not to bias nighttime fluxes towards too large CO2 emission.*

*We will add this reference and information to the manuscript.*

**#Has the data been filtered for u\*? Has any of this large variability been taken into account in the Reco vs temp calculation for GPP uncertainty (something to think about in future if not?).**

*The data has unintentionally not been filtered fur u\*. We determined the threshold at ~0.2 m s$^{-1}$for $CO_2$ and used the same value for COS. After reanalyzing the data, we observed only minor changes within daytime values and the products of the flux partitioning +. Due to the filtering of the nighttime data some minor changes of stated values within the results part will have to be applied. The overall patterns of fluxes remain and the manuscript will not have to be reworded.*

**There is a little repetition with the Results and Discussion being separate. I wouldn't object if the authors decided to combine both and tightened the text up. But obviously that's just a suggestion.**

*We thank reviewer 3 for the advice but prefer to keep the sections separated.*

**Minor comments by line number:**

**14: soil flux**

*We will change this according to the reviewer comment.*

**31: do you mean relative uptake? COS is in ppt vs CO2 in ppm**

*Yes, we will reword the sentence to more accurately correspond to the cited paper (Montzka 2007).*

**38: Extra bracket**

*We will add a comma and remove the bracket.*

**86: What kind of fertilizer (dairy? beef? pig?)? And when was it fertilized previously? Before the winter?**

*The grassland is fertilized with solid manure and cattle slurry (see Hörtnagl 2018) once a year at the end of the growing season in October. We will add the information to the manuscript:*

*Each year, the field site was fertilized with organic solid manure and slurry (Hörtnagl et al 2018) at the end of the season (07.10. in 2015).*

**140: Ambient COS from what height? There is a massive COS gradient so this will be important.**

*The intake height was at 0.12m above the ground and thus within the canopy. This information will be included in the method section.*

**160: I think this needs more explanation. What does an OBB represent? Is that good? Not good? If you aren't going into enough detail for readers to evaluate the model, then cut it. It's kind of hanging there with not enough info. And most of the packages mentioned will represent some mathematical approach to data analysis. Since packages come and go, it would be really helpful to have a sentence or two about what these packages actually represent.**

*The OOB score can be interpreted as a pseudo-R2 and is widely used in random forest analyses (regression and classification), especially in the absence of a proper test dataset. It uses the data not seen by the trees (random forest uses bootstrapping) as a test dataset. We will add this information to the methods section.*

**168: What heights along the tower were the gradients sampled from? How often were they sampled vs eddy flux sampling?**

*The air was sampled at 0.02m, 0.05m, 0.1m, 0.20m, 0.3m, 0.4m, 0.5m and 2.5m for 1 minute at each height at 2,2 slpm and at 1 Hz sampling frequency, compared to the eddy sampling frequency of 10 Hz. We will add this information to the methods section.*

**173: Was the eddy flux data filtered for insufficient turbulence? If so, what u\* filter was applied? How was the u\* threshold quantified? A plot of the FCOS and FCO2 vs u\* would be helpful here to understand the micro met dynamics for the site.**

*The u\* threshold was determined by running the change point detection algorithm of Barr et al (2013) on nighttime NEE. The u\* for the $CO_2$ flux (~0.2 m $s^{-1}$) was then applied for COS. We also tried to determine the u\* threshold for COS, but a satisfying change point couldn't be determined.*

*We noticed that the eddy flux data was unintentionally not filtered for u\* in the plots (which almost exclusively has only an effect during the night). We will update the plots and the corresponding values in the text.*

*The data has also been filtered for stationarity, integral turbulence test and footprint, as detailed by Gerdel. We also filtered for extreme values (< -100 pmol m$^{-2}$ s$^{-1}$ & > 100 pmol m$^{-2}$ s$^{-1}$).*

*We will add the plots of the FCO2 vs u\* to the supplement:*

[Figure]

**329: What does the [CO2] drop down to? Is there a relationship between u*/turbulence and the d[COS] and d[CO2]? That would be an interesting figure to see.**

*The CO2 mixing ratio drops down to 339 ppm at 0.1m above ground at 10 a.m. We will add a plot containing the u\* values and the differences of the $CO_2$ and COS mixing ratios between canopy level (0.4m) and 0.02 m for COS and 0,1m for $CO_2$ to the supplement. The two lowest measurement heights were excluded for $CO_2$ since the $CO_2$ mixing ratio increased due to the soil respiration.*

[Figure]

**422: How long does the morning increase in COS last for? Do you start to see a decrease in COS as the daytime uptake influences the air in the valley? Other sites have also seen this morning peak in COS. Maybe include a reference to those here. (e.g. Redwoods, Harvard Forest, etc)**

*We observed a steep morning increase in COS mixing ratios until about 11 a.m. (see attached figure). We will include this plot in the supplement and add the requested information to the discussion.*

---

## Author Response (AR1)

**A. Point by Point Response to Reviews**

Dear anonymous reviewers,

Thank you for your thorough reviews and your support in improve this manuscript.

**Response to Reviewer 1**

**Line 31. Are you sure that "the summer drawdown for COS is 6 times stronger than for CO2"? The magnitude of the times seems too large to be believable.**

*The exact wording of the cited paper Montzka et al. (2007) is:*

*'However, while reduced mixing ratios of CO2 during the NH growing season represent a balance between vegetative uptake and total respiration (i.e. NEP), the percentage reduction on COS mixing ratio is 4-6 times (5.5+/- 1.6) larger during June- August (calculated relative to mixing ratios measured at 4-8 km asl) (Figure 6c).'*

*We reworded the sentence to more accurately correspond to the cited paper (Montzka 2007):*

However, the relative decrease in ambient mixing ratio during summer of the northern hemisphere is 6 times stronger for COS than for $CO_2$ (Montzka et al., 2007) as COS is generally not emitted by plants like $CO_2$, which is released in respiration processes.

**Line 130. "while air was sucked through the chamber to the QCL at a flow rate of 1.5 l min-1". The heights of air inlets for the chamber and ambient environment should be noted because remarkable vertical distribution of COS mixing ratio near the ground was observed in this study. If the height of air inlet for the chamber was within the canopy of the grass, the COS uptake flux would be largely overestimated, e.g., the COS mixing ratio could drop to 134ppt within the canopy in comparison with about 500ppt over the canopy.**

*The intake height was at 0.12 m above the ground and thus within the canopy. The COS concentration inside the chamber was thus similar to what the undisturbed soil would experience, which avoids uptake/release being biased high/low when COS-enriched air from above the canopy would be used.*

*We included this information in the method section.*

*The intake height of the ambient as well as the inlet of the chamber air were located at 0.12 m above the ground and thus within the canopy height with the exception of right after the cuts (see cutting dates in Section 2.1).*

*However, we also have to disagree with the comment on the overestimation of the uptake flux if the intake was within the canopy. In order not to bias measurements, the mixing ratios used for chamber flux measurements should be as close to reality as possible, which is why air from within the canopy was used. Had we used COS-enriched air from above the canopy, any soil uptake would have been overestimated (because the COS gradient across the soil surface is increased), while any COS emission would have been underestimated (because the COS gradient across the soil surface is reduced).*

*We added this information to the discussion:*

The low COS mixing ratios observed in the lowermost canopy layers just above the soil surface emphasize the importance of using air from within the canopy for soil chamber measurements and not COS richer air from above the canopy, which would increase the COS gradient and thus increase uptake/decrease emission of COS to/from the soil.

**Line 228. What's the plant available water? Fig. 1 only presents the SWC (%) which is below 38% during almost all days, rather than 21 days.**

*The SWC in Fig.1 was replaced with the plant available water, which falls below 50 % during 111 days.*

**Line 248. "During nighttime (RSW = 0, n = 43), the soils of the grassland acted as a net sink for COS 74.4 % of the time" is better replaced by "During nighttime (RSW = 0, n = 43), 74.4 % of the COS emission fluxes were negative, implying soils of the grassland acted as a net sink for COS".**

*The sentence was changed as suggested.*

**Line 263. Why did you use both circles and open diamonds for depicting COS soil fluxes? What's the difference**
**between them?**

*We removed the depiction about the open diamonds, which were not present in the plots.*

**Lines 276-278. "Especially after the cuts we observed a strong decline in COS uptake and even times where the**
**grassland turned into a net source for COS with midday means of up to 24.5 pmol m-2s-1 (Fig. 4 b) for up to 8 days**
**after the cut, when the dried litter had already been removed (Fig. 2 a-c)". This sentence is suggested to be replaced**
**by "Especially after the cuts we observed a strong decline in COS uptake ((Fig. 4 b)) and the grassland even turned**
**into a net source for COS in middays (Fig. 2 a-c) with a highest emission flux of 24.5 pmol m-2s-1 in August after the**
**cut.".**

*We replaced the sentence according to your suggestion. To keep the crucial information about the grassland turning into a*
*net source for up to 8 days after the cut, we added an additional sentence to the manuscript:*

*We observed COS emissions for up to 8 days after the cut, when the dried litter had already been removed (Fig. 2 a-c).*

**Lines 280-281. "The cut in October led to a reduction in COS uptake, which was lowest three days after the cut (Fig.**
**2d)". The description seems to be inconsistent with the Fig. 2d.**

*We agree that the lowest COS uptake did occur later than 3 days after the cut. However, this is also related to the overall*
*decline in COS uptake by the grassland at the end of the season. We see no recovery of the COS flux after the last cut. We*
*rephrased this in the manuscript and included more data points to Fig. 2 d):*

*The cut in October led to a reduction in COS uptake, which declined across several days and did not recover, as the end of*
*the season was reached (Fig. 2 d & Fig. 5 b).*

**Lines 297-298. I don't understand the meaning of the sentence. Fig. 4a is the seasonal cycle of CO2, rather than COS.**

*We changed Fig 4a to 4c and added Fig 4a to subsequent sentence dealing with the seasonal response of respiration.*

**Lines 325-328. I wonder why the COS mixing ratio dropped so large during the nighttime when the COS uptake was**
**much less than that during midday.**

*Compared to the constant influx of COS rich air during daytime, due to the increased boundary layer (see line 422), this*
*influx stops during nighttime and COS gets depleted within the canopy, even when the COS uptake of the ecosystem is lower*
*than during daytime. The strong input of COS rich air during daytime has also been reported by other studies.*

*We added references to the manuscript (Campbell 2017 & Rastogi 2018).*

**Lines 375- 377. I don't understand the logic of this sentence. Because the chamber enclosed both soil and the residual**
**grass after the cuts, the COS emission under sunlight irradiation might be due to the residual rather than the soil**
**itself, e.g., the photochemical formation of COS from the possible liquid released from the cut grasses (JGR, 109,**
**D13301, doi:10.1029/2003JD004206, 2004; JES, 5 1 ( 2 0 1 7 ) 1 4 6 – 1 5 6). If the COS emission was ascribed to soil,**
**the authors are suggested to verify it by using a flow tube method under dark and irradiation conditions.**

We removed this sentence.

**Line 413. Why did the lowest COS mixing ratio appear in winter when vegetation COS uptake is relatively low?**

*During winter, no strong emission fluxes are expected to originate from vegetation and soils. The mixing ratios rather*
*depend on the transport of COS enriched air from oceans, which are also highest in summer (see Montzka 2007).*

**Lines 419-421: The above sentences didn't mention the difference in concentrations during day and nighttime.**

*We added the sentence:*

*Even though the COS mixing ratio at the layer closest to the soil were higher during day than during nighttime, the absolute*

*decrease in COS was lower during nighttime due to partial stomatal closure (Kooijmans et al., 2017;Campbell et al., 2017).*

*The absolute difference in concentrations during day and nighttime originate from changes in the height of the planetary*

*boundary layer (PBL).*

**Lines 421-422. Considering the much stronger COS uptake by the grass in daytime than in nighttime, COS mixing**

**ratio above the canopy should decrease in daytime, rather than nighttime despite of the variation of PBL**

*Several studies (e.g. Rastogi 2018 – Ecosystem fluxes of carbonyl sulfide in an old-growth forest: temporal dynamics and*

*responses to diffuse radiation and heat waves) showed that the PBL is the main influence factor on sub-diurnal variability in*

*COS mixing ratio. The incomplete stomatal closure as well as the soil sink cause the nighttime decrease in mixing ratio as*

*there is no influx of COS rich air from the atmosphere. The stronger daytime drawdown can also be observed in the gradient*

*analysis as the decrease in COS mixing ratio, from to the canopy height down to the soil was higher during daytime (125*

*ppt) compared to the nighttime decrease (102 ppt).*

*This information is already present in the manuscript; see line 325-328 and 419-423.*

**Response to Reviewer 2**

**1. Definition of "LRU on ecosystem scale": note that most LRUs in the literature were derived from branch chamber**

**measurements, and were then used in the relationship between Fcos and Fco2 (Eq.1), with the implication/assumption**

**that LRUs derived from branch chamber measurements are representative of the entire canopy. Here the authors**

**infer the LRU (of the entire canopy) from ecosystem flux measurements. Please clarify this.**

*We added that the LRU was calculated using eddy fluxes without the need to use chambers to the method section:*

*Using the above stated method infers LRU solely on the basis of fluxes on ecosystem scale, whereas other studies typically*

*used branch/leaf chamber measurements (Yang et al., 2018) to determine the relationship between the COS and CO2 uptake*

*rates.*

**2. CO2 observations: IRGA CO2 measurements were used in the analyses. I believe that the QCL also measured**

**CO2. Were those data used somehow? If IRGA CO2 measurements were calibrated to the WMO scale, CO2 should**

**be reported as mole fractions instead of mixing ratios, because the WMO scale (NOAA calibration gases) is reported**

**on mole fractions. The difference between mole fractions and mixing ratios is significant for CO2, and not significant**

**for COS.**

*The COS and $CO_2$ fluxes were calculated using solely the QCL data as stated in section 2.5.2. We followed the processing*

*steps of Gerdel et al. 2017 to retrieve the fluxes using the same filters, which as stated by Gerdel et al. 2017 has the*

*advantage that the influence of the high pass filter on the ecosystem relative uptake (ERU) largely cancels out, if applied on*

*COS as well as $CO_2$. The ambient COS and $CO_2$ concentrations both originated from the QCL data, which puts out mixing*

*ratios. We changed the method section accordingly since neither $CO_2$ nor $H_2O$ fluxes of the IRGA were used in the final*

*version of the manuscript. We apologize for the confusion.*

**3. What are the reasons for the relatively low enhancements of daily maximum PAR values reaching the soil surface**

**after the third and the fourth cuts (Figure 1)? These are not consistent with the "incident shortwave radiation**

**reaching the soil surface" in Figure 3e.**

*The data of the PAR reaching the soil surface in Fig 1 originated from a PAR sensor that was likely overgrown by short*

*vascular plants and mosses growing directly at the soil surface at the end of the season. We changed the data from this*

*sensor to the data of Fig 3e, which was calculated using the Beer-Lambert law (see line 151).*

**4. Fcosmedian turned to positive after the third cutting while remained largely negative after the fourth cutting**

**(Figure 2c&d), given that COS soil fluxes would be both positive. What could explain the difference here?**

*The modelled soil fluxes were always relatively small compared to the ecosystem scale fluxes and shouldn't be the reason for*

*the difference between fig 2c&d. Also, there is less incoming solar energy at the end of the season, likely also decreasing the*

*emission strength of the residual litter.*

*We added a sentence containing this to the discussion:*

*We did not observe strong COS emissions after the last cut, as the incoming solar radiation, which we hypothesize to amplify*

*the degradation of sulfur containing compounds of plants, was reduced at the end of the season.*

**5. High-light conditions: what is the definition of high-light conditions? How sensitive is the estimated LRU at high**

**light intensity to the choice of high-light conditions?**

*The parameter "iota" – LRU under high light conditions results from equation 8. The second parameter "kappa" controls*

*the exponential decrease of LRU when the incoming photosynthetic active radiation (PAR) is decreasing and limiting GPP*

*but not the COS flux.*

$$LRU = \iota \, e^{\left(\frac{\kappa}{R_{PAR}}\right)}$$

*While mathematically iota is only obtained at infinitely high PAR, in practice above about 700 µmol m$^{-2}$ s$^{-1}$ PAR only*

*insignificant change in the ecosystem relative uptake, reflecting the relationship between the COS and the $CO_2$ flux, can be*

*observed.*

*We included the definition for high light into the methods part:*

*While mathematically ι is only obtained at infinitely high PAR, in practice above about 700 µmol m$^{-1}$ s$^{-1}$ PAR (Kooijmans et*

*al., 2019) only insignificant change is reported in other studies (Stimler et al., 2011).*

**Other technical comments:**

**Line 111: I think it is more likely by a GC-MS than a GC, please double check.**

*We changed GC to GC-MS within the revised document.*

**L154: The unit of RSW-soil should be Wm-2, and for other places as well.**

*We changed this according to the reviewer comment.*

**L165: obtain-high resolution ! obtain high-resolution**

*We changed this according to the reviewer comment.*

**L191: Eq.7 was developed in earlier studies, please refer to the original work.**

*We changed this according to the reviewer comment and added (Sandoval-Soto 2005) as reference.*

**L198-203: It will read better if these are moved to after L188.**

*We changed this according to the reviewer comment.*

**L230: It needs a bit more explanation of NDVI, what does it indicate?**

*We changed the manuscript accordingly.*

**Figure 3 caption. open diamonds?**

*We removed the text part about the open diamonds, which are not present in the figure.*

**L312: why is an increase in RECO expected?**

*Even though there is a reduction in plant respiration, the increase in incoming radiation reaching the soil surface leads to*

*an increase in soil temperature and consequently soil respiration (see Fig.5a). We added this information to the manuscript:*

*While the grassland acted as a net sink for CO2 during periods of high LAI (Fig. 5 6 b), a combination of a decline in GPP*

*and an increase in daytime RECO, as more incoming radiation was heating the soil surface, turned it into a net source*

*during midday in periods of low LAI (Fig. 5 6 a).*

**L319: should be COS instead of CO2**

*We changed this according to the reviewer comment.*

**L433-435: LRU is a normalized ratio, and should not depend on the ambient COS. I do not get the point here.**

*This is not quite right. LRU is calculated in order to normalize for differences in COS (and $CO_2$) concentrations, which*

*affect the fluxes. For the same COS and CO2 flux and the same CO2 concentration, LRU will differ whether the ambient*

*COS concentration is 400 or 500 ppt. This is what we quantified in the linear perturbation analysis and what this sentence*

*refers to.*

**L437-439: Please specify which are the exact "those observations". Figure 4 indicates that low COS fluxes took place**

**shortly after the cuttings, which coincides with COS emissions from soils after the cuttings.**

*We clarified this by changing the sentence to:*

*For the calculation of LRUs we had to remove the canopy flux data containing COS and/or CO2 emissions observations*

*since these would yield negative values for ERU and LRU (see Eq.8).*

**L419-422: It may be worth pointing out that the vertical gradient of COS between the canopy level and below the**

**canopy levels exists throughout the day and night, but that of CO2 does not.**

*We added the information to the discussion.*

*We only observed an increase in CO2 mixing ratios, caused by the release of CO2 through respiration processes in the soil,*

*whereas COS mixing ratios further declined down to the soil surface.*

Response to Reviewer 3

**Minor comments in general:**

**There seems to be a really strong gradient within the grass canopy. Would the really low COS above the soils (100-**

**200 ppt) influence the COS flux?**

*Yes, since the exchange across the soil surface is driven by the concentration gradient between the ambient air just above*

*the soil surface and within the soil. We added a sentence containing this information to the discussion:*

*The low COS mixing ratios observed in the lowermost canopy layers just above the soil surface emphasize the importance of*

*using air from within the canopy for soil chamber measurements and not COS richer air from above the canopy, which*

*would increase the COS gradient and thus increase uptake/decrease emission of COS to/from the soil.*

**Out of interest, what does the FCOS/[COS] (COS deposition velocity) look like?**

*We provide the plot in the revised supplement.*

**I also think the concentration discussion (Sections 3.4, Fig 6, 4.3) should come before the flux discussion. It really sets**

**the context to fully appreciate the flux discussion.**

*We agree and moved the parts accordingly.*

**Data needs to be made public before publication! Make sure in the final version that the text in the figures is big**

**enough. I was having to zoom in a lot to read things.**

*The data is online now and the font size of the text within the figures was increased.*

**I'm really impressed at how well the FP+ model works for grass (Fig 5b/d).**

*Thank you, we were also very happy with the mean diel fluxes resulting from the model.*

**What drives the large change in CO2 variability between day and night?**

*As shown by Wohlfahrt et al. (2005), the large variability of NEE during nighttime conditions is due to the combination of*

*low wind speeds and stable stratification which results in highly intermittent $CO_2$ fluxes compared to well-mixed convective*

*daytime conditions. On a half-hourly basis, fluxes may even be negative (i.e. net uptake of $CO_2$), which is biologically*

*impossible, but results from the intermittent nature of the $CO_2$ transport and is typically compensated for by large emission*

*fluxes in a subsequent averaging period. As recommended by Wohlfahrt et al. (2005), $CO_2$ fluxes were filtered for u\*, but not*

*for the sign of the fluxes in order not to bias nighttime fluxes towards too large $CO_2$ emission.*

*We added this reference and information to the manuscript.*

**#Has the data been filtered for u*? Has any of this large variability been taken into account in the Reco vs temp**
**calculation for GPP uncertainty (something to think about in future if not?).**

*The data has unintentionally not been filtered for u\*. We determined the threshold at ~0.2 m s$^{-1}$for $CO_2$ and used the same*
*value for COS. After reanalyzing the data, we observed only minor changes and no changes in the overall patterns. Text and*
*figures were adapted accordingly. We attached all plots before and after the correction at the end of this document. During*
*the reanalysis we were also able to recover more data from immediately after the first cut, which slightly increased LRU and*
*ERU during this phase in Fig. 7a.*

**There is a little repetition with the Results and Discussion being separate. I wouldn't object if the authors decided to**
**combine both and tightened the text up. But obviously that's just a suggestion.**

*We thank reviewer 3 for the advice but prefer to keep the sections separated.*

*We removed several redundancies.*

**Minor comments by line number:**

**14: soil flux**

*We changed this according to the reviewer comment.*

**31: do you mean relative uptake? COS is in ppt vs CO2 in ppm**

*Yes, we reworded the sentence to more accurately correspond to the cited paper (Montzka 2007):*

*However, the relative decrease in ambient mixing ratio during summer of the northern hemisphere is 6 times stronger for*
*COS than for CO2, (Montzka et al., 2007) as COS is generally not emitted by plants like CO2, which is released in*
*respiration processes.*

**38: Extra bracket**

*We added a comma and removed the bracket.*

**86: What kind of fertilizer (dairy? beef? pig?)? And when was it fertilized previously? Before the winter?**

*The grassland is fertilized with solid manure and cattle slurry (see Hörtnagl et al. 2018) once a year at the end of the*
*growing season in October. We added the information to the manuscript:*

*Each year, the field site was fertilized with solid manure and cattle slurry (Hörtnagl et al., 2018) at the end of the season*
*(07.10. in 2015).*

**140: Ambient COS from what height? There is a massive COS gradient so this will be important.**

*The intake height was at 0.12m above the ground and thus within the canopy with the exception of measurements taken just*
*after the cuts. This information is now included in the method section:*

*The intake height of the ambient as well as the inlet of the chamber air were located at 0.12 m above the ground and thus*
*within the canopy height with the exception of measurements right after the cuts (see cutting dates in Section 2.1).*

**160: I think this needs more explanation. What does an OBB represent? Is that good? Not good? If you aren't going**
**into enough detail for readers to evaluate the model, then cut it. It's kind of hanging there with not enough info. And**
**most of the packages mentioned will represent some mathematical approach to data analysis. Since packages come**
**and go, it would be really helpful to have a sentence or two about what these packages actually represent.**

*The OOB score can be interpreted as a pseudo-R2 and is widely used in random forest analyses (regression and*
*classification), especially in the absence of a proper test dataset. It uses the data not seen by the trees (random forest uses*
*bootstrapping) as a test dataset. We added this information to the methods section.*

**168: What heights along the tower were the gradients sampled from? How often were they sampled vs eddy flux**
**sampling?**

*The information was already present in the methods section. See 2.3 and 2.3.1*

**173: Was the eddy flux data filtered for insufficient turbulence? If so, what u\* filter was applied? How was the u\***
**threshold quantified? A plot of the FCOS and FCO2 vs u\* would be helpful here to understand the micro met**
**dynamics for the site.**
*The u\* threshold was determined by running the change point detection algorithm of Barr et al (2013) on nighttime NEE.*
*The u\* for the $CO_2$ flux (~0.2 m $s^{-1}$) was then applied for COS. We also tried to determine the u\* threshold for COS, but a*
*satisfying change point couldn't be determined.*
*We noticed that the eddy flux data was unintentionally not correctly filtered for u\* in the plots (which almost exclusively has*
*only an effect during the night). The data in the plots and the corresponding values in the text have been updated.*
*We added the plot of the $FCO_2$ vs u\* to the supplement.*
**329: What does the [CO2] drop down to? Is there a relationship between u\*/turbulence and the d[COS] and d[CO2]?**
**That would be an interesting figure to see.**
*The $CO_2$ mixing ratio drops down to 339 ppm at 0.1m above ground at 10 a.m. We added a plot containing the u\* values*
*and the differences of the $CO_2$ and COS mixing ratios between canopy level (0.4m) and 0.02 m for COS and 0,1m for $CO_2$ to*
*the supplement. The two lowest measurement heights were excluded for $CO_2$ since there the $CO_2$ mixing ratio increased due*
*to the soil respiration.*
**422: How long does the morning increase in COS last for? Do you start to see a decrease in COS as the daytime**
**uptake influences the air in the valley? Other sites have also seen this morning peak in COS. Maybe include a**
**reference to those here. (e.g. Redwoods, Harvard Forest, etc)**
*We observed a steep morning increase in COS mixing ratios until about 11 a.m.. We included include this plot in the*
*supplement and added the requested information to the discussion.*
*While the PBL is shallow during nighttime and the COS mixing ratio decreases due to the sink strength of the grassland, at*
*the onset of the day, the PBL layer height increases quickly and COS rich air is transported down to the ecosystem* (see Fig.
S12) *(Campbell et al., 2017). A similar steep increase until midday has also been observed by Rastogi et al. (2018).*

*Updated figures:*

*New:*

[Figure]

*Old:*

[Figure]

*New:*

[Figure]

*Old:*

[Figure]

*New:*

[Figure]

*Old:*

[Figure]

*New:*

[Figure]

[Figure]

*Old:*

**B. List of relevant changes**

• The correct u* filter is now applied and all values in the document have been changed accordingly

• During the reanalysis we were also able to recover more data from immediately after the first cut, which slightly increased LRU and ERU during this phase (Fig. 7a)

• The section about the COS and CO2 mixing ratios is now placed before the flux sections in the result as well as the discussion section

[revised manuscript text omitted]

---

## Author Response (AR2)

**Point by Point Response to Reviews**

Dear anonymous reviewers, thank you for your thorough reviews and your support in improving this manuscript.

**R1 commented:**

The authors properly respond to the comments raised by the reviewers and made corresponding revision in the text. However, there are still several mistakes or improper places in the revised manuscript, e.g., line 111, 113, 126, the digital dates are easily confused with values, especially for line 113 "before the 16.06" which is better present as 16 June;

**We agree and changed the section accordingly.**

Line 114-115, the sentence lack of logic and the values in "the 16. and the 18.6.." are typos;

**We restructured the sentence and present the dates in a different style "June 16 and July 18".**

Lines 132-137, the size of the chamber should be noted, rather than a reference;

**We added the chamber volume (4155 cm³) to the methods section.**

The difference of the air temperatures in the chamber and the ambient air is better noted to make sure the flux measurements are reliable;

**Due to the interaction of temperatures sensors and COS, we don't have chamber temperatures available. Therefore we stated the ambient and soil temperatures.**

The method used for COS flux measurements may have technical problem by using the difference of COS mixing ratios between those measured in the chamber and above the chamber because the chamber inlet was located at 0.12m above the ground where COS mixing ratio was usually much lower than those measured above the chamber. The relatively low COS mixing ratios in the chamber is not attributed to the uptake by the chamber covered area. To exactly obtain COS flux, COS mixing ratios at the inlet of the chamber and in the chamber are needed.

**The inlet of the air flowing into the chamber and the inlet of the ambient air are placed at the exact same height and beside each other. There was no difference in mixing ratios between the ambient and the chamber inlet air.**

Additionally, the relevant references are suggested to be cited to support your statement in lines 63-65, e.g., the influence of soil temperature and moisture on soil COS emission (Liu et. al., 2010, Bio geosciences 7(2):753-762).

**We added the reference to the manuscript.**

**R2 commented:**

The only thing I can't seem to find is where they say what the value of the u* filter is for CO2. It's mentioned in the comments but doesn't seem to have been included in the manuscript. Once they add that, I would recommend accepting as is.

**We added the information to the methods section.**

**R3 commented:**

Only a technical correction is needed. The fact that QCL outputs mixing ratios does not justify its correctness, as stated in the responses "The ambient COS and CO2 concentrations both originated from the QCL data, which puts out mixing ratios". As commented earlier, if the measurements were calibrated to the WMO scale, CO2 should be reported as mole fractions instead of mixing ratios, because the WMO scale (NOAA calibration gases) is reported on mole fractions.

**We agree with the reviewer. All instances of mixing ratio within the document (including the figures) have been replaced with mole fraction.**

**Relevant changes**

- All instances of mixing ratio were changed to mole fraction
- We added the size of the soil chamber and the u* threshold to the document

**Tracked document**

[revised manuscript text omitted]